# A DNA demethylase reduces seed size by decreasing the DNA methylation of AT-rich transposable elements in soybean
Wanpeng Wang [1,6,7], Tianxu Zhang[2,3,7], Chunyu Liu[4,7], Chunyan Liu [5], Zhenfeng Jiang[5], Zhaohan Zhang[1], Shahid Ali[1], Zhuozheng Li[1], Jiang Wang[1,3], Shanwen Sun[1,3], Qingshan Chen[5,8] ✉, Qingzhu Zhang [2,3,4,8] ✉ & Linan Xie[1,3,8] ✉

Understanding how to increase soybean yield is crucial for global food security. The genetic and epigenetic factors influencing seed size, a major crop yield determinant, are not fully understood. We explore the role of DNA demethylase GmDMEa in soybean seed size. Our research indicates that GmDMEa negatively correlates with soybean seed size. Using CRISPR-Cas9, we edited *GmDMEa* in the Dongnong soybean cultivar, known for small seeds. Modified plants had larger seeds and greater yields without altering plant architecture or seed nutrition. GmDMEa preferentially demethylates AT-rich transposable elements, thus activating genes and transcription factors associated with the abscisic acid pathway, which typically decreases seed size. Chromosomal substitution lines confirm that these modifications are inheritable, suggesting a stable epigenetic method to boost seed size in future breeding. Our findings provide insights into epigenetic seed size control and suggest a strategy for improving crop yields through the epigenetic regulation of crucial genes. This work implies that targeted epigenetic modification has practical agricultural applications, potentially enhancing food production without compromising crop quality.

Soybean [*Glycine max* (L.) Merr.] is the most commonly cultivated seed legume in the world and provides protein and vegetable oil for livestock feed, human consumption, and industrial production[1]. Seed yield is a crucial trait in crop breeding and is influenced by multiple quantitative trait loci and environmental factors. Semidwarf cultivars of rice and wheat show improved fertilizer responses, lodging resistance, and light utilization, resulting in increased yields owing to their compact size, better biomass potential, and sturdier stems supports grains[2]. The remarkable rapid increase in maize yield can be attributed to increases in plant height, modification of the leaf angle, and exploitation of hybrid vigour[3,4]. However, yield formation in legumes is different from that in cereals. As soybean is a typical pod crop, if plant shape is modified, such as by semidwarfing, the number of pods will be significantly reduced and negatively impact soybean yields[5]. The yield of soybean crops depends on the weight and number of seeds per unit area, which are influenced by several factors, such as plant height, internode formation, numbers of branches, pods per plant, seeds per pod, and seed size[5]. Currently, one effective breeding strategy is to enhance the seed weight/size of varieties that are suitable for dense planting. This approach can help crops adjust to high-density planting conditions while maintaining a high photosynthetic efficiency, leading to improved seed masses and ultimately higher yields. Several cellular processes have been shown to regulate seed size by modulating maternal tissue growth, including the ubiquitin-proteasome pathway, G-protein signalling, mitogen-activated protein kinase (MAPK) signalling, phytohormone perception and homeostasis, and certain transcription factors (TFs)[6]. Despite significant progress in identifying genes and mechanisms that govern seed size, comprehensive quantitative trait loci (QTL) analyses have revealed that genetic variation accounts for only approximately 75% of seed size traits[7]. Therefore,

[1]Key Laboratory of Saline-alkali Vegetation Ecology Restoration, Ministry of Education, College of Life Science, Northeast Forestry University, Harbin, Heilongjiang, China. [2]State Key Laboratory of Tree Genetics and Breeding, Northeast Forestry University, Harbin, Heilongjiang, China. [3]The Center for Basic Forestry Research, College of Forestry, Northeast Forestry University, Harbin, Heilongjiang, China. [4]College of Life Science, Northeast Forestry University, Harbin, Heilongjiang, China. [5]College of Agriculture, Northeast Agricultural University, Harbin, Heilongjiang, China. [6]Present address: Qingdao Institute of Bioenergy and Bioprocess Technology, Chinese Academy of Sciences, Qingdao, Shandong, China. [7]These authors contributed equally: Wanpeng Wang, Tianxu Zhang, Chunyu Liu. [8]These authors jointly supervised this work: Qingshan Chen, Qingzhu Zhang, Linan Xie. ✉e-mail: qshchen@126.com; qingzhu.zhang@nefu.edu.cn; linanxie@nefu.edu.cn

additional crucial mechanisms that contribute to seed size regulation remain to be elucidated.

One piece of epigenetic machinery can control the whole-genome chromatin state, thereby potentially affecting the activity of multiple proximal protein-coding genes. For example, RNA-directed DNA methylation (RdDM) modulates the expression of various downstream genes, including *OsMIR156d/j* and *D14*, by influencing the methylation of miniature inverted-repeat transposable elements (MITEs). This mechanism ultimately impacts rice tillering[8]. Plants establish cytosine methylation in CG, CHG, and CHH contexts via the RdDM pathway. DNA methylation is maintained by specific enzymes and is dynamically regulated through balanced methylation and demethylation processes[9–13]. During soybean seed development, CHH methylation undergoes significant dynamic changes, primarily targeting transposable elements (TEs). Hypomethylated regions within the soybean genome contain critical genes involved in seed development and transcriptional regulation[14,15]. These findings suggest that CHH-hypomethylation may play a critical role in regulating seed development. *DEMETER* (DME) is a bifunctional DNA glycosylase/lyase that mediates active DNA demethylation in *Arabidopsis*. It is favourably expressed in the central cell of the female gametophyte and in seeds[16–20]. Active DNA demethylation mediated by DME preferentially targets small, AT-rich, and nucleosome-depleted euchromatic TEs[21]. In *Arabidopsis*, loss-of-function mutants of DME (*dme-2*) exhibit larger seeds but also experience seed abortion and abnormal cell proliferation[22,23]. While DNA demethylases play a crucial role in the development of crop seeds, there is a limited understanding of the precise mechanism by which DME affects the size of soybean seeds[24–26].

In this study, we investigated the differences in gene expression and the methylation levels of CG, CHG, and CHH context in soybean varieties with large and small seeds. Our results suggest that the level of CHH-demethylation, which is regulated by *GmDMEa*, plays a significant role in determining the size and weight of soybean seeds. We introduced knock-out mutations into *GmDMEa* and subsequently observed a considerable increase in soybean seed size, primarily due to an enlargement of cell size, resulting in a higher 100-seed weight and plant yield. Remarkably, the observed increase in soybean seed size did not result in any observable abnormal growth or developmental phenotypes throughout the soybean life cycle. Our study also revealed significant increases in CHH methylation in dry seeds upon *GmDMEa* loss of function, in which MuDR transposons were preferentially targeted, and decreases in the expression of several known genes involved in the ABA response, transcription factors (TFs), and ubiquitin pathway genes that control seed size. Additionally, we employed chromosome segment substitution line (CSSL) progeny to confirm that *GmDMEa*-controlled soybean seed size and weight could be inherited.

## Results

### DME negatively controls soybean seed size

We collected 11 different soybean germplasms, which included three wild soybean germplasms (*G. soja*): ZYD00006 (ZYD6), Baiyangdian (Bai YD), and Dongying (Dong Y); two USA cultivars with genomic sequences: Williams 82 (W82) and Jack; and six randomly selected cultivars from Northeast China: Dongnong 50 (DN50), Dongnong 60 (DN60), Dongnong 42 (DN42), Hefeng 51 (HF51), Hefeng 55 (HF55), and Suinong 14 (SN14). We divided them into two groups based on seed size: a small-seed group and a large-seed group. To assess seed size, we measured the length, width, and thickness of seeds, and we evaluated the 100-seed weight of the dry seeds (Fig. 1a, b).

To investigate global gene expression patterns in the small- and large-seed groups, we conducted an RNA-seq analysis using dry seeds and identified differentially expressed genes (DEGs) based on the thresholds of a fold change >2 and -$\log_{10}$FDR > 2. There were 490 upregulated genes and 768 downregulated genes in the large-seed group compared to the small-seed group (Supplementary Data 1).

To evaluate the potential biological functions of the DEGs, Gene Ontology (GO) analysis was performed on the DEGs (Fig. 1c,

Supplementary Fig. 1a). The top GO terms of the downregulated DEGs included "Transcription regulator activity" (GO:0140110, 84 genes; *p*-value, 4.44E-04), "Reproduction" (GO:0003700, 82 genes; *p*-value, 5.30E-04), "Embryo development" (GO:0009790, 39 genes; *p*-value, 0.0019), and "Post-embryonic development" (GO:0009791, 110 genes; *p*-value, 0.0021) (Supplementary Data 2). Genes associated with these terms are closely related to seed development. Twenty-one genes were related to "Epigenetic, regulation of gene expression" (GO:0040029) in the GO enrichment analysis of downregulated genes (Fig. 1c, Supplementary Data 2). RNA-dependent RNA polymerase 2 (*GmRDR2, Glyma.05G035900, Glyma.17G091500*) and demethylase (*GmDMEa, Glyma.10G202200*) were significantly downregulated in the large-seed group, which suggests that DNA methylation control by epigenetic factors plays a critical role in the regulation of seed size in soybean (Fig. 1d, e; Supplementary Fig. 1b). However, the upregulated genes were not enriched in functions related to seed development.

The seed size-related downregulation of *GmRDR2* and *GmDME* described above prompted us to investigate DNA methylation level differences between small and large seeds. Whole-genome bisulfite sequencing (WGBS) of dry seeds was used to profile the methylomes of dry seeds from the eleven soybean germplasms. Overall, each methylome was sequenced with >30-fold coverage per strand, and more than 90% of the genomic cytosine positions were covered in each sample.

To obtain an overview of the DNA methylation profile, we compared the methylation levels of three different sequence contexts (CG, CHG, and CHH) in gene regions, intergenic regions, and TE regions. Our results showed that DNA methylation levels were significantly different between the large and small-seed groups, particularly in the CHH context in TE regions. Specifically, we identified 13,097 hypo-differentially methylated regions in CHH contexts (CHH hypo-DMRs) and 21,753 hyper-differentially methylated regions in CHH contexts (CHH hyper-DMRs) in the large-seed group, whereas the number of CG/CHG hypo- and hyper-DMRs was much lower. (Fig. 1f, g and Supplementary Data 3).

The proportion of TEs in CHH hyper-DMRs (18%) was higher than that in CHH hypo-DMRs, suggesting that increased CHH methylation in TE regions is related to larger seed size (Fig. 1h). We further analysed the CHH methylation levels of the top three TE types (Gypsy, Copia, and MuDR) in the soybean genome among the eleven soybean germplasms (Supplementary Fig. 2a). The results showed that the CHH methylation levels of all TE types were higher in the large-seed group than in the small-seed group, suggesting that high CHH methylation levels in TEs are associated with larger seed size (Fig. 1i and Supplementary Fig. 2b).

To investigate the role of DNA demethylases in regulating seed size in soybean, we analysed the demethylase family members present in soybean (*Glycine max*), alfalfa (*Medicago truncatula*), *Arabidopsis thaliana* and five other species. There are two members of the demethylase family in soybean, GmDMEa (*Glyma.10g202200*) and GmDMEb (*Glyma.20g188300*), with GmDMEa showing higher homology to AtDME (*AT5G04560*) and MtrDME (*Medtr1g492760*) (Supplementary Fig. 3a). Both GmDMEs contain an RNA recognition motif (RRM), Perm-CXXC unit, and ENDO3c domain, consistent with the typical structure of a demethylase (Supplementary Fig. 3b, Supplementary Data 4). In the comparison of these amino acid sequences, GmDMEa showed higher similarity (34.57%) to AtDME (Supplementary Data 5). We further examined the expression patterns of *GmDMEs* in various soybean tissues, including root, epicotyl, hypocotyl, unifoliate leaf, shoot meristem, flower, and the 1st, 3rd, 5th, 7th, and 9th trifoliolate leaves of DN50 plants. Our results showed that GmDMEs were expressed at the highest levels in dry seeds, with *Glyma.10G202200* showing specific expression in DN50 dry seeds (Supplementary Fig. 3c). In contrast, other demethylases in soybean, such as GmROS1a (*Glyma.03g190800*), GmROS1b (*Glyma.10g065900*), and GmROS1c (*Glyma.13g151000*), did not show specific expression in dry seeds. Upon comparing the expression levels of *GmDMEa* and *GmDMEb* in groups with large and small soybean seeds, we observed a significant difference in the expression of *GmDMEa* between the two groups, while *GmDMEb* showed no significant

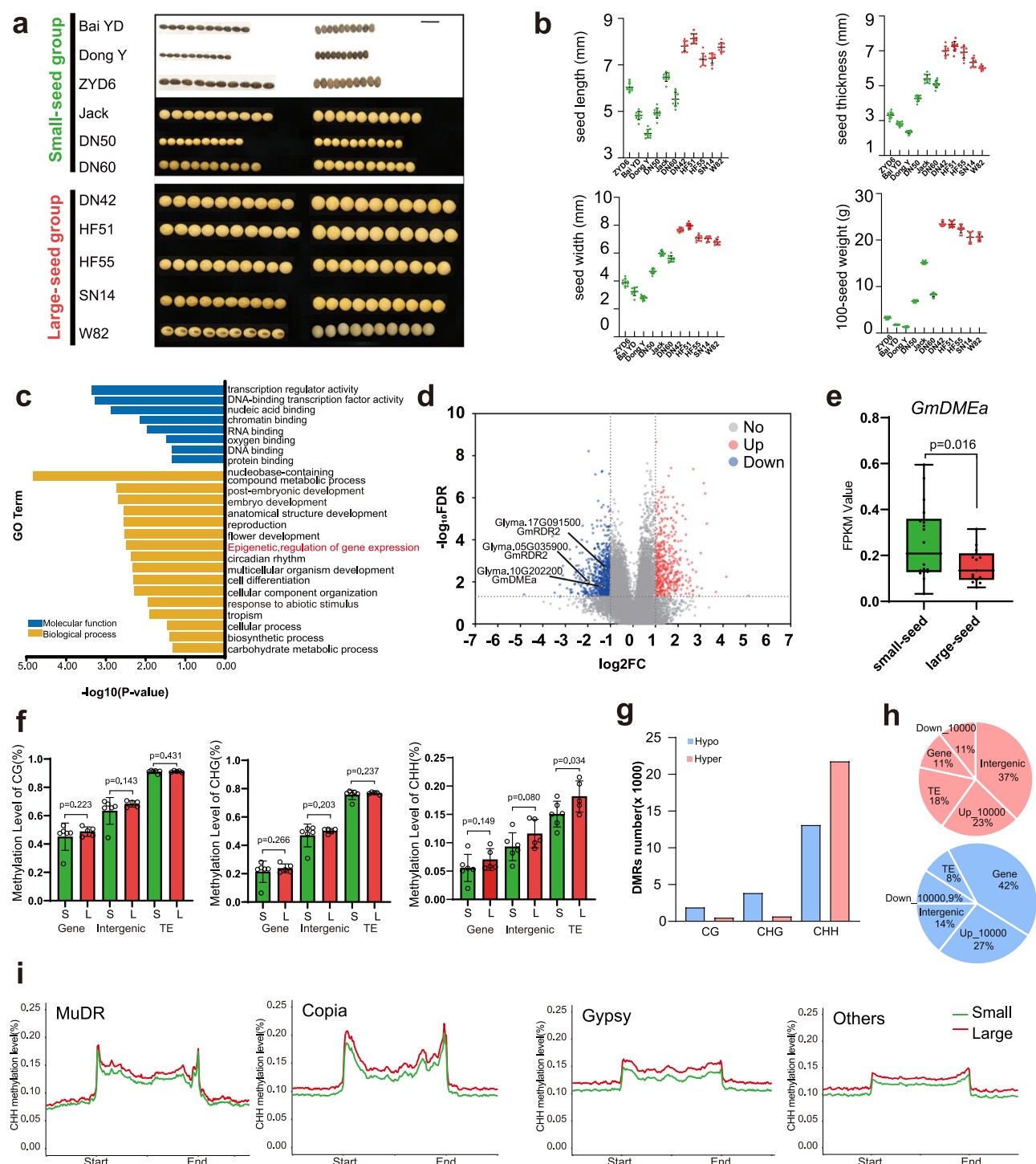

variation(Fig. 1e; Supplementary Fig. 1b). Consequently, we propose that the DNA demethylase GmDMEa is a key epigenetic regulator of soybean seed size.

### *gmdmea* mutants have larger seeds

To study the role of *GmDMEa* in the regulation of gene expression and control of seed size in soybean, we employed the CRISPR/Cas9 approach to create mutations in *GmDMEa* in the background of DN50, a dense-planting cultivar. We designed single guide RNAs (sgRNAs) identical to the nucleotide sequences of the first exon of *GmDMEa* to generate mutations corresponding to seven mutants (Fig. 2a, Supplementary Fig. 3a, b, and Supplementary Data 6). To validate these mutations, we performed PCR

amplification with primers containing cleaved amplified polymorphic sequence (CAPS) markers, followed by restriction enzyme digestion (Fig. 2b and Supplementary Fig. 4, 11). Notably, all mutations, except for *gmdmea-6* and *gmdmea-7*, lacked the *Sfa*NI recognition site but contained the *Xcm*I recognition site. Conversely, the mutations in *gmdmea-6* and *gmdmea-7* had the *Sfa*NI recognition site but lost the *Xcm*I recognition site. As a result, the PCR fragment from DN50 could be digested into a 346 bp fragment and a 180 bp fragment by *Sfa*NI, whereas the *gmdmea-1, gmdmea-2, gmdmea-3, gmdmea-4, and gmdmea-5* mutants could not be digested by *Sfa*NI. Similarly, the PCR fragment from DN50 could be digested into a 346 bp fragment and a 180 bp fragment by *Xcm*I, whereas the *gmdmea-6* and *gmdmea-7* mutants could not be digested by *Xcm*I.

**Fig. 1 | Expression and methylation patterns in dry soybean seeds. a** Eleven soybean ecotypes were categorized into small-seed (depicted in green) and large-seed (depicted in red) groups based on seed dimensions, specifically length, width, and thickness. Scale bar represents 1 cm. **b** Measurements of length, width, thickness, and weight for 100 dry seeds were recorded. Data represent mean values with standard deviations (SDs) for at least 10 seeds from each ecotype ($n = 10$). Small-seed and large-seed groups are indicated by green and red points, respectively. **c** Gene Ontology (GO) enrichment analysis for downregulated genes between the two groups is shown, with molecular functions marked in blue and biological processes in yellow. The negative logarithm of the enrichment $p$-value visualizes the significance of differential expression. **d** A volcano plot illustrates the differentially expressed genes (DEGs) between the groups. Genes with a fold change exceeding 2 and an FDR below 0.01 are emphasized. Upregulated genes are marked with red dots, downregulated genes with blue, and *GmDMEa* and *GmRDR2* with yellow. **e** Transcriptional variation of *GmDMEa* in dry soybean seeds. Transcript levels, quantified as FPKM, were derived from transcriptomic analyses of dry seeds. Data are presented in a box plot representation, where the interquartile range is delineated by the box, the median is indicated by the horizontal line within the box, and whiskers extend from the minimum to the maximum values. The small-seed cohort

($n = 18$; green box) includes 'DN50', 'ZYD6', 'DN60', 'Dong Y', 'Bai YD', and 'Jack', while the large-seed cohort ($n = 15$; red box) comprises 'W82', 'SN14', 'HF51', 'DN42', and 'HF55'. Statistical significance was assessed by one-tailed, unpaired Student's t-tests with a 95% confidence interval. Differences were considered statistically significant at $p < 0.05$. **f** Differential methylation patterns in dry soybean seeds. Methylation levels were quantified at CG, CHG, and CHH nucleotide contexts across gene bodies, intergenic regions, and transposable elements (TEs) within the soybean genome. The bar graph contrasts the methylation percentages between large (L) and small (S) seed groups. Differences were considered statistically significant at $p < 0.05$. **g** A stacked bar chart displays the counts of differentially methylated regions (DMRs) contrasting the large- and small-seed groups. Hypo-DMRs (lower methylation in the large-seed group) are shown in blue, and hyper-DMRs (higher methylation in the large-seed group) in orange. **h** A pie chart depicts the distribution of CHH-DMRs across the soybean genome, with Up_10000 and Down_10000 denoting regions 10,000 base pairs upstream and downstream from genes, respectively. **i** Average CHH methylation levels across four types of TEs in both small and large-seed groups are plotted in a line graph, with the small-seed group in green and the large-seed group in red.

The *gmdmea-1*, *gmdmea-3*, *gmdmea-5*, *gmdmea-6*, and *gmdmea-7* mutants produced a frameshift and a premature stop codon, leading to a truncated polypeptide missing amino acids at the C-terminal end of GmDMEA in DN50. *gmdmea-2* and *gmdmea-4* had deletions of 1 or 2 amino acid in the N-terminus of GmDMEA (Supplementary Data 6). Seeds from homozygous *gmdmea* mutants in the T2 generation were used for the subsequent experiments.

To further investigate the function of *GmDMEa* in determining seed size, we conducted a phenotypic analysis of all homozygous mutants, focusing on traits related to yield in particular. Seed length, width, and the thickness of dry seeds were measured to assess seed size. Compared with DN50, the *gmdmea* mutants showed a significant increase in seed size and the 100-seed weight (Fig. 2c, d; Supplementary Fig 5a). Additionally, we collected seeds from each plant to determine the yield per plant, which showed a significant increase in the mutants (Fig. 2d; Supplementary Fig. 5a). However, we observed no significant differences in the seed oil or protein content (percentage of seed weight) between the mutants and DN50 (Fig. 2e). The number of pods and branches per plant showed no significant difference between the mutants and DN50 (Fig. 2f; Supplementary Fig. 5). Hence, the *gmdmea* mutants exhibited a significant enhancement of their seed size and weight phenotypes, without any changes in plant architecture or nutrient content.

Among all of the mutants, the most remarkable increase in yield was observed in *gmdmea-3*, and we subsequently conducted further investigations using homozygous loss-of-function mutants of *gmdmea-3*. Considering the potential adverse effects on plant growth and reproduction associated with DME mutations[22,23], we examined the phenotypes of the *gmdmea-3* mutant during growth and seed development stages. No significant difference at plant height or architecture was observed at different growth stages from R1 (Reproductive growth stage 1, stage at which the soybean begins flowering) to R8 (Reproductive growth stage 8, the final stage of soybean growth, where 95% of the pods have changed to their mature brown color) (Fig. 2g and Supplementary Fig. 5c, 6). More importantly, no obvious abortive seeds were found during the entire reproductive period (Fig. 2h–l). We specifically examined seed and pod development and measured the seed size of *gmdmea-3* mutants in different stages according to days after flowering (DAF), including the early pod development, developing, mature and harvestable stages. Seed size showed significant differences in the mutants throughout all growth and development stages (Fig. 2m).

**The *gmdmea-3* mutant has a larger seed size due to increased cell size**

To investigate the cellular phenotype associated with the seed size increase of the *gmdmea-3* mutant, we utilized transverse and longitudinal sections of mature seeds to observe the cell size and cell number within each section.

Half of the seeds were sectioned at three positions perpendicular to the cotyledon separation (transversally) and two positions parallel to the cotyledon separation (longitudinally) for frozen sectioning to describe the cellular morphology of whole seeds (Fig. 3a–c). Thirty fixed area fields of view at each position were randomly selected, and the cell number in each field of view was counted via microscopy, after which the average cell size at each position was calculated. Subsequently, we calculated the average cell number in each section based on the entire section area and average cell size. Our results indicated that the average cell size of the *gmdmea-3* mutant at each position was significantly larger than that in the DN50 control (Fig. 3d). However, the average cell number at each position did not exhibit a significant difference (Fig. 3e). Our findings suggest that the increase in seed size in the *gmdmea-3* mutant is attributed to enhanced cell size.

**DME-mediated DNA demethylation regulates gene expression by modulating the methylation levels of AT-rich TEs**

To investigate the role of *GmDMEa* in DNA demethylation associated with seed size, WGBS was used to establish the methylome profiles of dry seeds from DN50 and *gmdmea-3* plants. Then, we analysed the distribution of DNA methylation along gene bodies, TEs, and their flanking regions (1 kb upstream and downstream). Our findings revealed that although global CG and CHG methylation levels remained unchanged, the CHH methylation level was significantly elevated in the bodies of TEs but not in gene bodies or flanking regions in *gmdmea-3* (Fig. 4a, b, Supplementary Fig. 7a). We identified a total of 33,645 CHH hyper-DMRs and 9,064 CHH hypo-DMRs, which were more abundant than CG and CHG hypo-DMRs and hyper-DMRs (Fig. 4c; Supplementary Data 7). Our results confirmed that GmDME impacts DNA demethylation, especially CHH methylation, in soybean seeds.

To further analyse the distribution of CHH hyper-DMRs, we classified TE types in the soybean genome according to TE numbers (Supplementary Fig. 2a). Our results revealed that CHH hyper-DMRs were highly enriched in MuDRs (Fig. 4d), even though they accounted for only 10.06% of all TEs, which is lower than the percentages of Gypsy and Copia TEs in the soybean genome (Supplementary Fig. 2a). We characterized all TEs in the soybean genome according to the AT ratio and found that MuDRs presented the highest AT-ratio among all TE types (Fig. 4e; Supplementary Fig. 7a). Furthermore, the analysis of the distances between TEs and genes showed that MuDRs were located closer to genes than the other TE types (Fig. 4f), which implies that GmDMEa might target AT-rich TEs to control gene expression.

To assess the effects of CHH methylation on gene expression, we performed RNA-seq using the dry seeds of DN50 and *gmdmea-3* plants and identified 918 downregulated genes and 473 upregulated DEGs (Fig. 4g, Supplementary Data 8). Subsequent screening for genes with CHH-DMRs within the upstream 10 kb regions revealed 7358 CHH hyper-DMR-

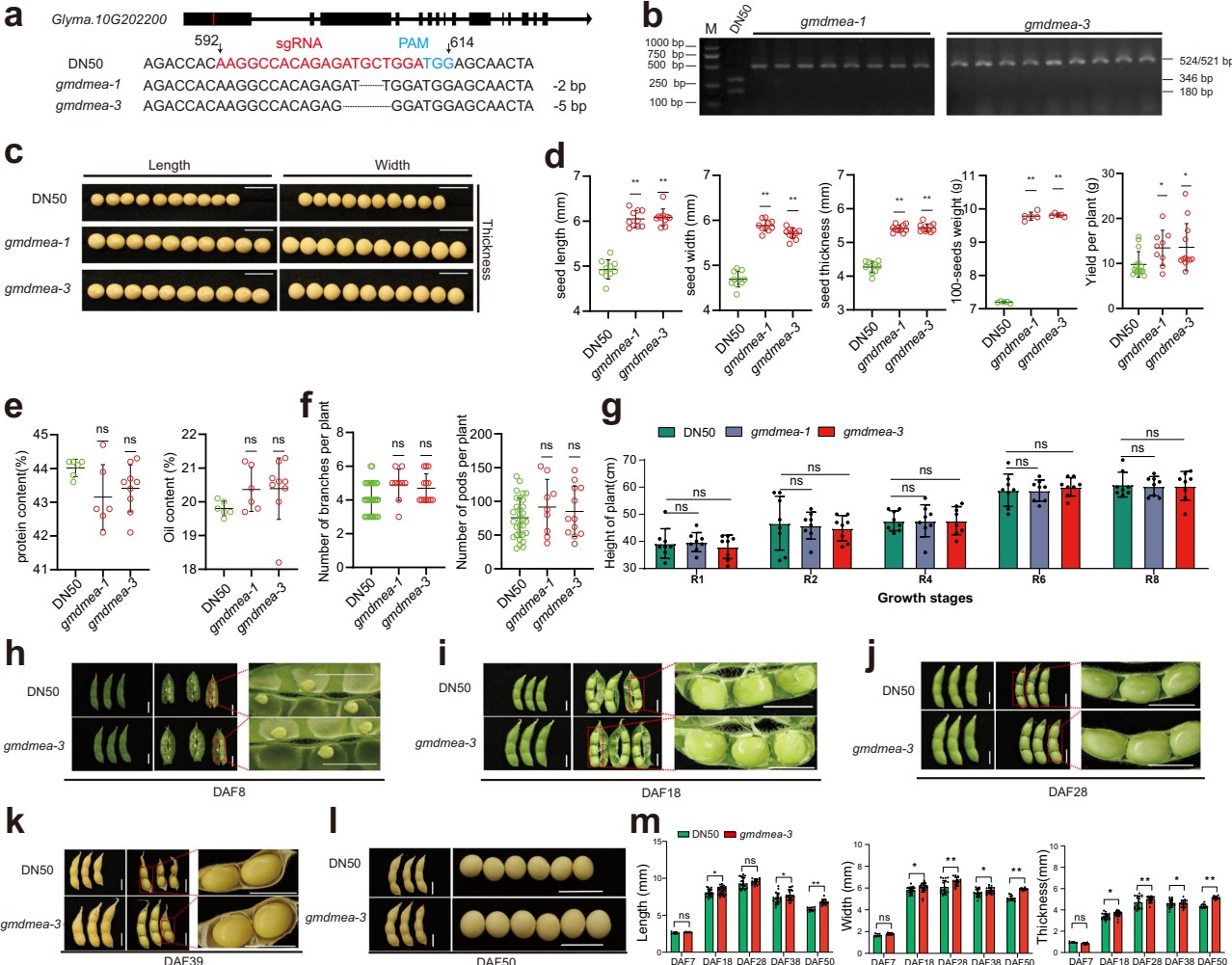

**Fig. 2 | Identification and phenotype of gmdmea mutants. a** Generation of mutations in *GmDMEa* (*Glyma.10G202200*) by the CRISPR/Cas9 method using single-guide RNA (sgRNA). The sgRNA sequence is shown in red, and the proto-spacer adjacent motif (PAM) sequence is shown in blue. The coding sequence of *GmDMEa* that is the same as the sgRNA and its flanking sequences are compared with the corresponding sequences in the *gmdmea-1* and *gmdmea-3* mutants. The numbers (592 and 614) above the nucleotide sequences are relative to the start codon of *GmDMEa*. **b** CAPS markers were used to identify the mutations in the *gmdmea* mutants. PCR products amplified by specific primers from DN50, *gmdmea-1* and *gmdmea-3* were digested by *SfaN I*. The primers used in this experiment are listed in Data 9. M represents the DNA ladder. **c** Photos of mature seeds harvested from DN50, *gmdmea-1* and *gmdmea-3* plants to show seed length, width, and thickness. Scale Bar = 1 cm. **d** Seed size and per-plant yield for DN50, *gmdmea-1*, and *gmdmea-3* genotypes. Seed size data are expressed as mean ± SD from at least 10 individual seeds per biological replicate. The 100-seed weight is also shown as mean ± SD, based on quintuplicate measurements. **e** Protein and oil composition in dry seeds from

DN50, *gmdmea-1*, and *gmdmea-3* plants. Each value represents the mean ± SD of six biological replicates. Pod and branch counts per plant for each line (DN50, *gmdmea-1*, and *gmdmea-3*) are also presented. **f** Pod and branch counts per plant for in DN50, *gmdmea-1*, and *gmdmea-3* lines. Data are mean ± SD for a minimum of nine biological replicates. **g** Plant height trajectory across reproductive stages R1 to R8. 'R' denotes the reproductive phase, with stages R1 to R8 encompassing pod maturation to physiological maturity, culminating in harvest readiness. Displayed are mean values ± SD, with a minimum of nine biological replicates. **h–l** Complete pods, developing seeds in pods and magnified photo of DN50 and *gmdmea-3* seeds at different development stages. DAF, days after flowering. Bar = 1 cm. **m** Seed dimensional metrics (length, width, and thickness) at multiple developmental phases. Data represent mean ± SD for no fewer than nine biological replicates. All statistical disparities between means were evaluated using unpaired two-tailed t-tests, signifying significance with *$p < 0.05$ (significant), **$p < 0.01$ (highly significant), ns $p > 0.05$ (not significant).

associated genes and 2475 CHH hypo-DMR-associated genes in the soybean genome. Among the downregulated genes in the *gmdmea-3* mutant, we identified 49 CHH-hyper-DEGs. (Fig. 4g, Supplementary Data 9). In examining the involvement of transposable elements (TEs), we found that 38 of these 49 downregulated CHH-hyper-DEGs had upstream TEs, with a total of 119 TEs. Particularly noteworthy is the proportion of MuDRs; 46 MuDR TEs were identified, accounting for 39.66% of TEs upstream of the CHH-hyper-DEGs. This is a marked enrichment compared to the whole genome, where out of 55,589 genes, 19,853 genes have upstream TEs comprising a total of 160,990 TEs, with MuDR TEs making up 23.96% of this total (Fig. 4h; Supplementary Fig. 7c). Such significant enrichment of

MuDRs upstream of CHH-hyper-DEGs underscores their potential influence on gene expression regulation, particularly in the context of genes downregulated in the *gmdmea-3* mutant. These results reveal that GmDMEa regulates gene expression by affecting the methylation levels of TEs, especially MuDRs.

## Decreased expression of ABA-responsive genes and transcription factors in *gmdmea-3*

In this study, we aimed to identify genes associated with seed size by comparing the transcriptomes of wild soybean (*G. soja*) with those of cultivated soybeans. Given the presence of unique genes in wild soybeans, we

**Fig. 3 | Cell size and cell number in mature seeds of DN50 and gmdmea-3 mutants. a** Schematic diagram of the transverse and longitudinal sections of halves of mature seeds. T1, T2, and T3 are transverse sections from half seeds, where the seeds were divided into four parts evenly along the seed length. L1 and L2 are longitudinal sections from half seeds, where the seeds were divided into three parts evenly along the seed width. **b** Morphology of the transverse and longitudinal sections (left, bar = 500 μm) and (**c**) partial magnification (right, bar = 50 μm) of DN50 and *gmdmea-3*. **d** Average cell size in sections of mature seeds of DN50 and *gmdmea-3*. Values are means ± SDs (*n* = 30). **e** Average cell number of each section calculated according to the total area of the section and average cell size. Values are means ± SDs (*n* = 5). Statistical disparities between means were evaluated using unpaired two-tailed t-tests, signifying significance with *\*p* < 0.05 (significant), *\*\*p* < 0.01 (highly significant), ns *p* > 0.05 (not significant).

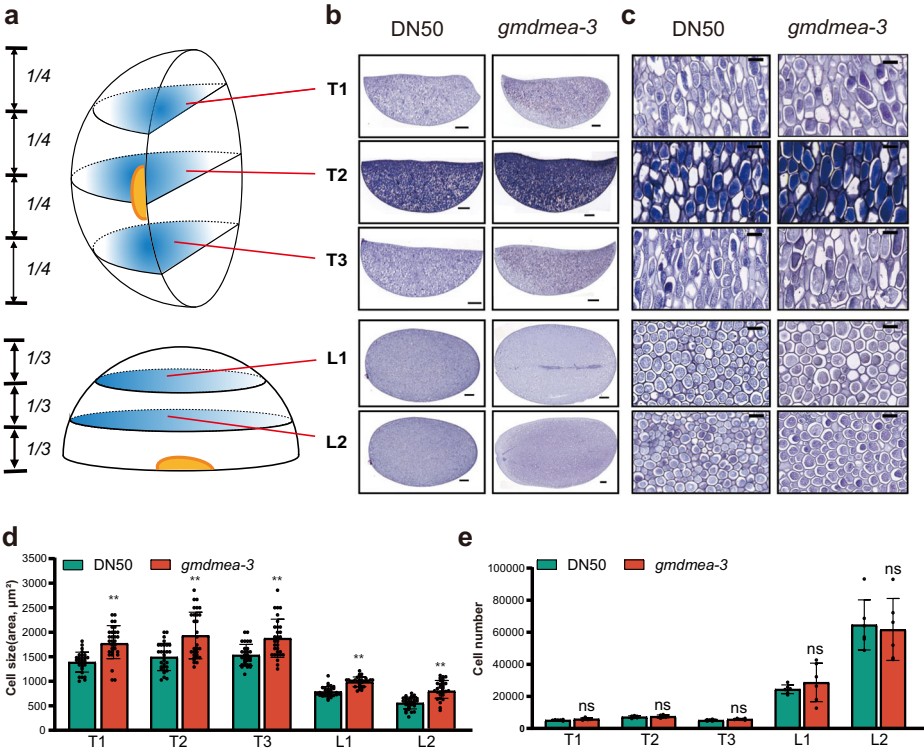

first conducted comparisons among three wild soybean ecotypes to exclude genes that are unique to the wild varieties. Subsequently, we compared these genes with small-seed cultivated varieties to remove DEGs unrelated to seed size, ultimately identifying 413 DEGs potentially related to the small seed phenotype. Additionally, we compared wild soybeans with large-seed cultivars, resulting in 266 DEGs, and further analysis yielded 200 internal DEGs within wild soybeans. Since these 200 genes were identified in wild soybeans with similar seed sizes, they are likely associated with traits other than seed size.

Further, by cross-comparing these three sets of DEGs, we excluded the unique effects of wild soybeans, resulting in 322 candidate DEGs related to small seed size. We also identified 6643 DEGs between the small-seed cultivars DN50 and DN60, which were categorized as unrelated to seed size. By removing these non-seed size related genes from the DEGs between gmdmea-3 and DN50, we pinpointed 892 DEGs closely associated with seed size, including 569 downregulated genes and 323 upregulated genes. (Supplementary Fig. 8, Supplementary Data 10).

GO analysis of the 569 downregulated DEGs revealed that the enriched functional terms were related to the regulation of transcription, with 71 genes functioning in "DNA-binding transcription factor activity" (GO: 0003700, corrected Padj = 0.0097) and 72 genes functioning in "transcription regulator activity" (GO: 0140110, Padj = 0.0058). Additionally, 63 genes were related to "response to lipid" (GO: 0033993 Padj = 0.0015), and 42 genes were related to "response to abscisic acid" (GO: 0009737 Padj = 0.0019) (Fig. 5a). We further analysed the GO-enriched TFs and ABA response genes. The upregulated expressed genes did not show significant GO enrichment (Supplementary Data 11).

To identify the genes directly affected by DME-mediated demethylation that were associated with the aforementioned functional genes, we conducted a comparison of transcription factors (TFs) and abscisic acid (ABA) response genes with the 49 CHH-hyper-DEGs and generated the corresponding protein-interaction (PPI) network. The amino acid sequences were uploaded to the STRING database website (https://cn.string-db.org/Version: 11.5), and some published protein interactions[27–29] were added based on previous reports. Some of these genes formed intricate and dense networks, indicating their interdependent functions (Fig. 5b). In

the PPI network, *Glyma.17G005600* (LEC1)[28], *Glyma.19G228000* (AIF4)[30], *Glyma.03G249000* (AN3)[31], and *Glyma.06G022300* (FD)[32] have been reported to play a role in seed development or to directly suppress seed size. We examined three CHH-hyper-DEGs: *Glyma.06G029100* (BLH3), *Glyma.20G094500* (GoLS1), and *Glyma.01g015700* (SAMBA). In our study, we focused on the promoter regions of three CHH-hyper-DEGs: *Glyma.06G029100* (BLH3), *Glyma.20G094500* (GoLS1), and *Glyma.01g015700* (SAMBA), using IGV snapshots to confirm the positions of these DNA methylation differentially methylated regions (DMRs) that also contained MuDR transposable elements. These DMRs were specifically targeted for McrBC-qPCR analysis to quantify DNA methylation levels (Fig. 5c), and corresponding gene expression was verified by RT-qPCR. The primers used for the McrBC-qPCR and RT-qPCR analyses are detailed in Supplementary Data 14. The results from McrBC-qPCR and RT-qPCR confirmed the DNA methylation status and gene expression levels (Fig. 5d, e).

Since ABA-responsive genes expression was significantly downregulated in *gmdmea-3*, we aimed to clarify the role of ABA in *gmdmea-3* seed enlargement by testing for differences in endogenous ABA contents in soybean seeds at different developmental stages. We performed ABA extraction and content determination referring to previously reported methods[33,34] (see Materials and Methods). The results showed that the ABA content in the seeds temporarily increased during the early development of soybean seeds from DAF10-DAF18 and gradually decreased after DAF18. However, there was no significant difference in the endogenous ABA content between DN50 and *gmdmea-3* at any seed developmental stage (Supplementary Fig. 9). Thus, we concluded that the *gmdmea-3* mutant seeds became larger because of ABA-responsive gene downregulation, rather than a reduction in the endogenous ABA content.

## Using the DME marker to develop new soybean cultivars for yield improvement

To further demonstrate the heritability and universality of GmDMEa-mediated CHH-hyper associated genes related to the control of seed size, six progeny populations of the CSSL derived from a cross between SN14 and ZYD6 (specifically R183, R147, R102, R191, R178, and R84) were employed in the present study (Fig. 6a). In CSSL, the cultivated soybean variety SN14,

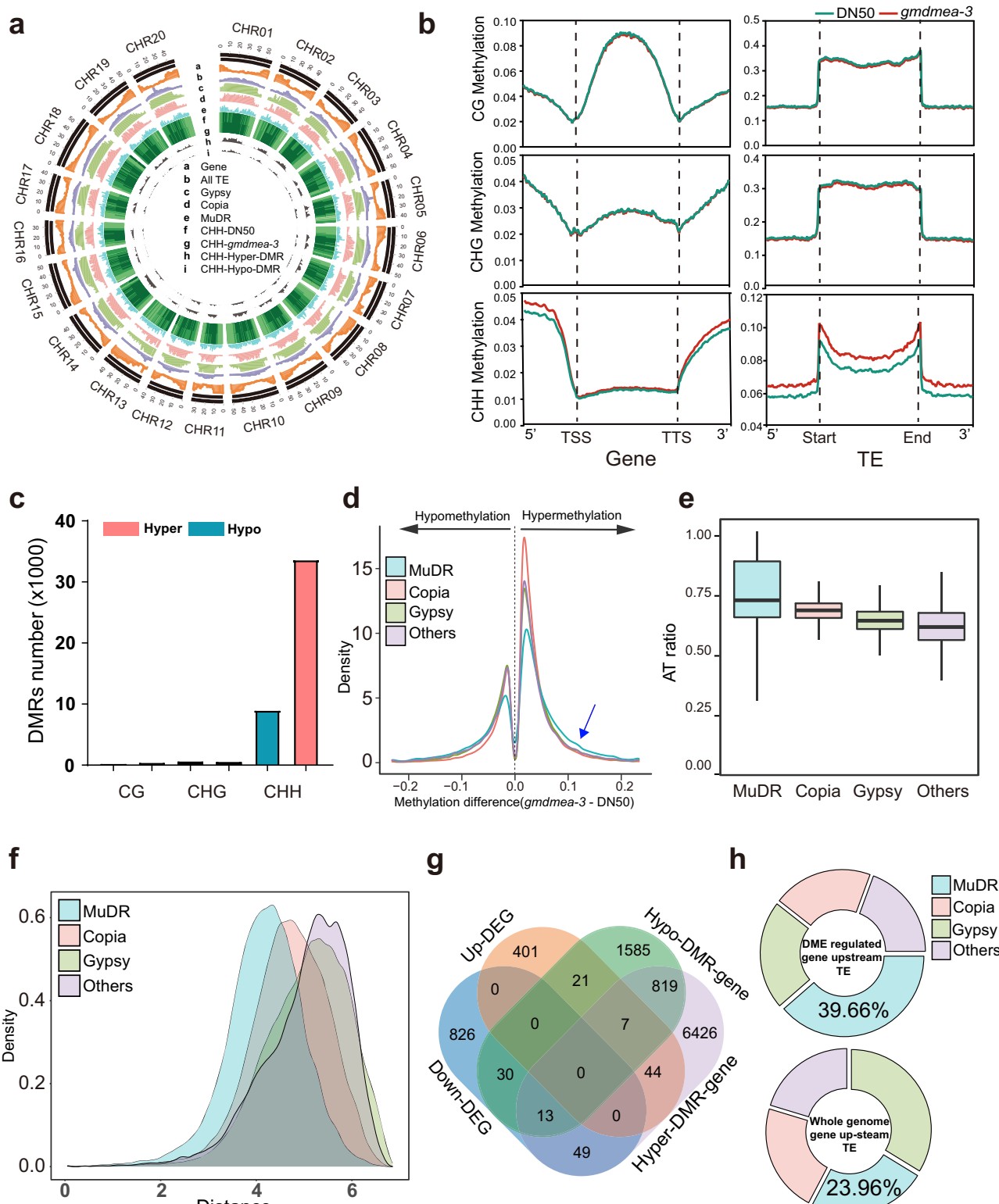

**Fig. 4 | In the MuDR transposon, CHH methylation suppresses gene expression.**
**a** Circos plot indicating the genome-wide distribution of genes, all TEs, Gypsy, Copia, MuDR, and CHH-DMRs in wild type (DN50) and the mutant (*gmdmea-3*). **b** Methylation levels within gene, TE and 1 kb flanking (upstream and downstream) regions in CG, CHG and CHH sequence contexts of DN50 (green) and *gmdmea-3* (red). **c** Number of hypermethylated (red bar) or hypomethylated (green bar) differentially methylated regions (hyper-DMRs and hypo-DMRs) in CG, CHG and CHH sequence contexts. **d** CHH methylation level changes in the three major (Copia, Gypsy and MuDR) and other types of transposons, where transposons that did not show methylation changes (0.0) were removed. **e** AT ratios of different transposon types throughout the soybean whole genome. **f** Distances of MuDR, Copia, Gypsy and the other transposons from genes. The X-axis represents the distance to the gene, and the Y-axis represents the density of TEs. **g** Four-way Venn diagram indicating the number of DEGs and genes associated with CHH-DMR (hypergene and hypogene). **h** Distribution of transposons located upstream of DME-regulated DEGs (upper) and all genes of the soybean genome (lower).

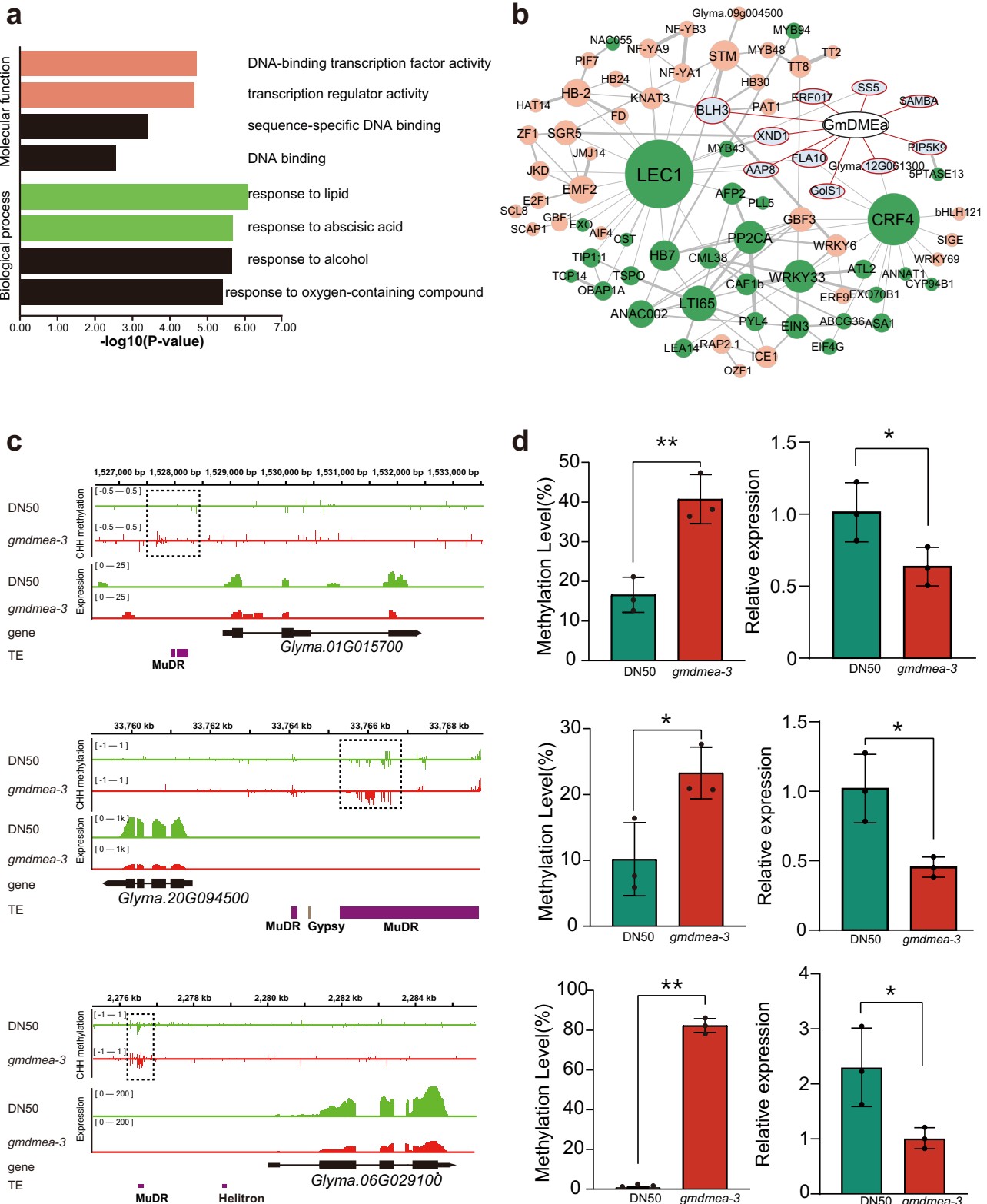

**Fig. 5 | ABA response genes, transcription factors, and ubiquitin pathway genes are downregulated due to GmDMEa mutation-induced hypermethylation. a** GO enrichment analysis of the DEGs between DN50 and the *gmdmea-3* mutant. The bars show the -log10 values. The threshold for DEGs was set at a fold change ≥2 and p adjusted <0.05. **b** The interactions among CHH-hyper-DEGs (dark green), and GO-enriched TFs (light red) and ABA response genes (light green) were revealed by the PPI network produced with STRING. **c** Snapshots from the Integrative Genomics Viewer (IGV) showing DNA methylation and gene expression levels of *SAMBA* (*Glyma.01g015700*), *BLH3* (*Glyma.06G029100*) and *GoLS1*

(*Glyma.20G094500*)in DN50 and *gmdmea-3*. Dashed frames indicated as the CHH-DMRs located in upstream of the three genes. The height of each column represents the methylation level at each cytosine. The lower part indicates the transcript level of *SAMBA*, *BLH3* and *GoLS1*. **d** McrBC-qRT-PCR (left) was used to measure the methylation levels of CHH-DMRs in the *SAMBA*, *BLH3* and *GoLS1* genes. Expression levels (right) of the above genes in *gmdmea-3* revealed by qRT-PCR. Data are shown as the mean ± SD (*n* = 3). Statistical disparities between means were evaluated using unpaired two-tailed t-tests, signifying significance with *$p < 0.05$ (significant), **$p < 0.01$ (highly significant).

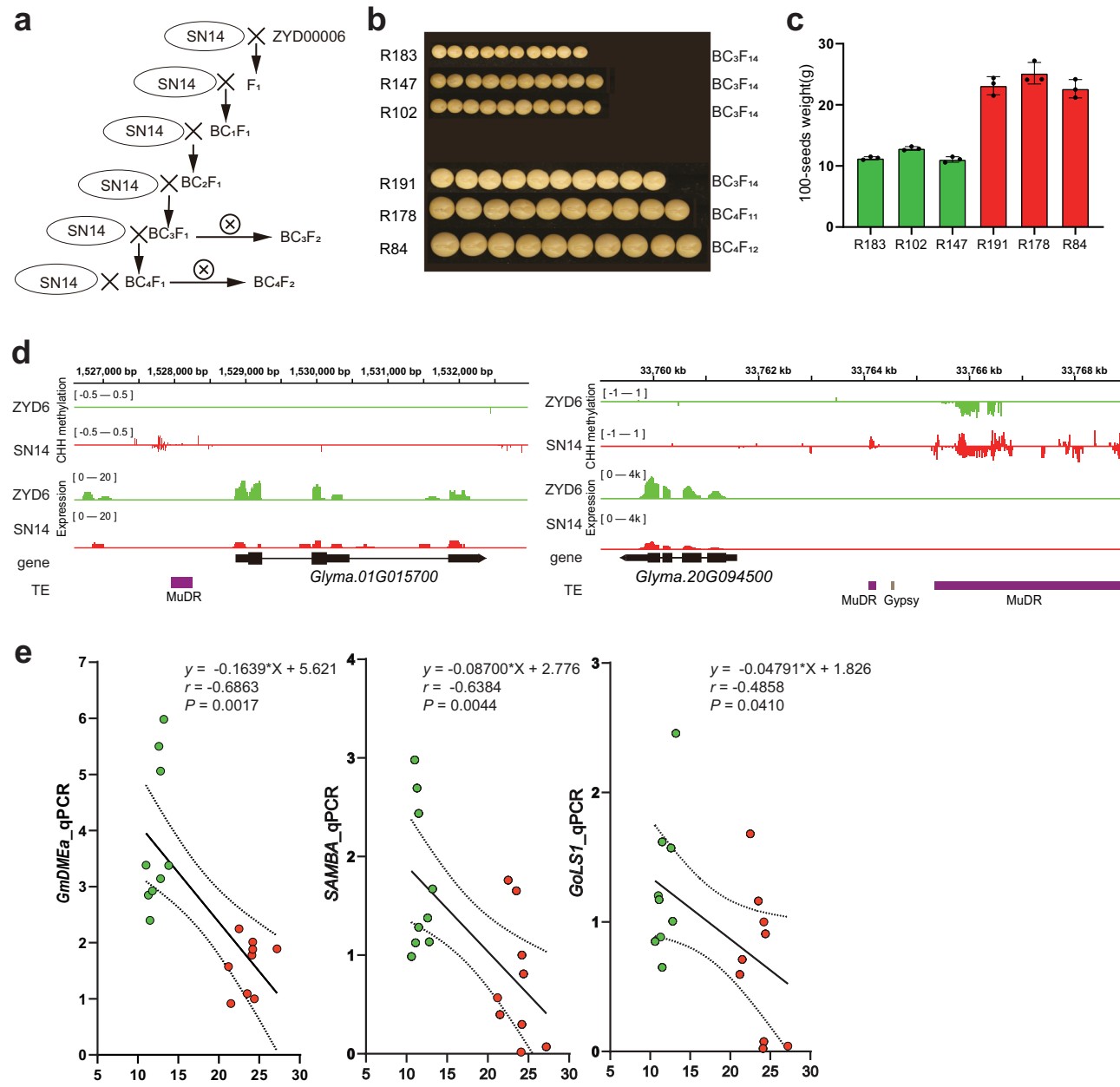

**Fig. 6 | In CSSLs, the suppression of seed size by DME is inherited across generations. a** Schematic diagram of the CSSL population construction procedure. **b** Dry seeds of CSSL progeny: R183, R147, R102 (small), R191, R178, and R84 (large).Bar = 1 cm. **c** 100-seed weight of CSSL progeny:R183, R147, R102 (small, with green bar), R191, R178, and R84 (large, with red bar). **d** Snapshots from the Integrative Genomics Viewer (IGV) showing DNA methylation and gene expression levels in *SAMBA* (*Glyma.01g015700*) and *GoLS1* (*Glyma.20G094500*) in dry seeds of SN14 and ZYD6. **e** Correlation of 100-seed weight with the expression levels of *GmDMEa*, *SAMBA* and *GoLS1* in progeny of CSSLs. The green dots represent small-sized seeds, while the red dots represent large-sized seeds. A total of six samples were analyzed, with three biological replicates for each sample.Pearson correlation test; *n* = 18.

characterized by large seeds, was used as the recurrent parent, while the wild soybean variety ZYD6, known for its small seeds, served as the donor. The seed sizes of the different CSSL progeny populations varied significantly, with the 100-seed weight and seed size showing significant differences between the small-seed progenies (R183, R147, R102) and large-seed progenies (R191, R178, R84) (Fig. 6b, c).

In the parents of CSSL, the expression level of *GmDMEa* was found to be low in SN14 and high in ZYD6 (Supplementary Fig. 10a). Furthermore, the GmDMEa-mediated CHH-hyper associated genes *SAMBA* and *GoLS1* showed higher DNA methylation levels in their promoter regions, especially in MuDRs, and lower expression levels in SN14. In contrast, ZYD6 displayed an opposite expression pattern (Fig. 6d). In the progeny of the CSSL,

the expression pattern of *GmDMEa* was consistent with that of the parents, with higher expression levels being observed in small-seed progenies and lower levels in large-seed progenies (Supplementary Fig. 10b). The associated analysis revealed a strong negative correlation between the expression levels of *GmDMEa*, *SAMBA*, and *GoLS1* and seed size (Fig. 6e; Supplementary Fig. 10b). In summary, the *GmDMEa* gene exhibits a stable genetic effect in regulating soybean seed size.

Taken together, our investigations revealed a pivotal role of GmDMEa in the regulation of soybean seed size by mediating CHH demethylation. GmDMEa preferentially targets AT-rich TEs and governs gene expression to exert control over soybean seed size, as represented in Fig. 7. The findings highlight the potential for the

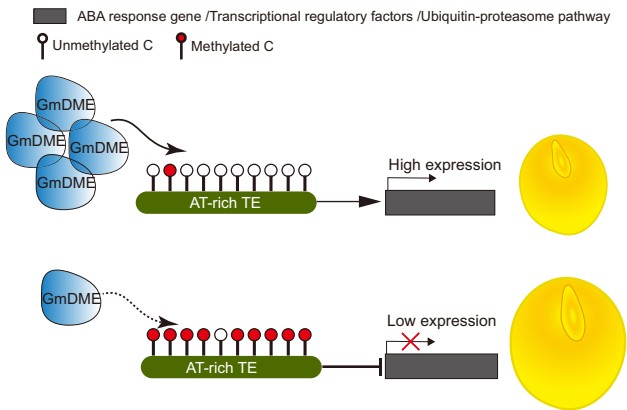

**Fig. 7 | Model of GmDME-mediated suppression of seed size in soybean.** GmDME performed active DNA demethylation at TEs located in promoter regions with a preference for AT-rich TEs and induces the expression of genes that negatively regulate seed size in soybean, thus producing small seeds.

manipulation of *GmDMEa* as a master gene for yield improvement in crops.

## Discussion

In our study, GmDMEa-mediated CHH demethylation played a role in suppressing soybean seed size. This finding was consistent with other studies showing that DNA active demethylation driven by demethylases is critical in plant development and crop traits. Changes in CHH methylation during seed development and some genes essential for seed development were located within hypomethylated regions of the soybean genome[14,15]. A reduction in *MtDME* expression leads to hypermethylation and downregulation of genes associated with nodule differentiation[35]. Mutations in the DNA demethylase OsROS1 result in a thickened aleurone and improved nutritional value in rice grains[36]. *DML3* is expressed at the onset of and during senescence when promoter regions, gene body regions, or 3' UTRs are demethylated to activate a set of *SAGs* to regulate leaf senescence[37]. A reduction in CHH methylation caused by OsNRPD1a and OsNRPD1b at MITEs in the promoters of *OsMIR156d* and *OsMIR156j* results in the derepression of these two genes, contributing to a high-tillering phenotype[8].

TE regions are targets of epigenetic factors. We found that GmDME preferentially targets AT-rich TEs, similar to results observed in *Arabidopsis*, in which AtDME mainly affects the methylation of AT-rich transposons[21]. In maize, *MDR1* (a *DME* gene homolog) directly regulates the methylation levels of Helitron TEs[38]. In soybean, although the content of TE-MuDR in the whole genome is only 10.06%, the content of AT in MuDR transposons is 73%, which is higher than that in any other transposon species in the whole genome (the AT proportion is 63%).

The DME-mediated CHH-hyper-DEGs were mainly ABA-response genes, a large proportion of which were transcription factors. Previous studies have shown that the ABA metabolic pathway and TFs significantly affect seed development and regulate seed size[39,40]. *BLH3*, which binds the important ABA-related gene *ABI3* and affects growth and development, is also affected[41]. Furthermore, several genes of the ubiquitin−proteasome pathway, like *SAMBA*, showed increased promoter methylation and contained MuDR TEs[42]. Although gene expression related to the ABA metabolic pathway and TFs significantly changed, there was no significant change in ABA content in the maturation stages. ABA contents changed dynamically during development, initially increasing in a short period during the development of soybean seeds, reaching a peak, and then gradually decreasing. These results confirmed and complemented previous findings in *Arabidopsis* (the dynamic changes in ABA content in the early stage of seed development were not reported)[43].

The *gmdme* mutants exhibits seeds with a significantly increased size, 100-seed weight, and yield per plant but normal protein and oil contents. Notably, there was no observable abnormal phenotype throughout seed development in this mutant, indicating the potential for breeding to obtain larger seeds while maintaining desirable traits in dense-planting cultivars. In contrast, *dme* mutants in *Arabidopsis* exhibit large seeds but suffer from high seed abortion rates, exceeding 90%[23].

These findings suggested that the cultivation pattern of the soybean variety DN50, which can be planted densely, was not altered and that only change was a seed size increase, without impacting nutritional quality. Combining dense planting with a larger seed size is a promising strategy for increasing soybean yields, and the development of large-seeded cultivars suitable for dense planting is expected to enhance the yields of existing cultivars.

While precise breeding using key genes identified through rational design or genome editing is an ideal breeding strategy, it is not uncommon for a single gene to control a trait, especially among developmental regulatory genes[44,45]. Our study demonstrates the crucial role of DME in soybean seed size control, highlighting its potential as a master gene capable of achieving "one-cause, multiple-effect" control of complex quantitative traits that can hopefully be applied in crop breeding and drive breakthrough improvements in yield.

Nevertheless, several scientific questions remain unresolved. One particular concern is the cause of differential *GmDMEa* expression between small and large-seed groups, the dynamic patterns of RdDM and DNA demethylation require further investigation. Additionally, while the *gmdmea-3* mutant exhibits a greater cell area, the number of cells does not differ significantly in the mutant. As larger soybean seeds often result from an increase in either the number or size of cells, studies suggest that *GmBBM*, *PP2C-1*, and *GmPSKγ1* act as positive regulators of seed growth[46–48], highlighting the potential for different mechanisms to be involved in the negative regulation of soybean seed size due to *GmDME* mutation. In the present study, we utilized a single replicate for our genome-wide DNA methylation sequencing. This approach was informed by a consideration of resources and precedents in the field where single replicates have been deemed acceptable in certain contexts, particularly given the relative stability of DNA methylation patterns in plants[49,50]. We acknowledge that this constitutes a limitation of our study, as biological replicates are the gold standard for omics research to ensure data reliability. To mitigate this limitation and substantiate the authenticity of our findings, we employed McrBC-qRT-PCR, a method with established accuracy for DNA methylation detection, as an additional validation step[51]. While the results from the McrBC-qRT-PCR experiments support the reliability of our sequencing data, we caution that the absence of biological replicates may affect the generalizability of our findings. Future research could address this limitation by including multiple biological replicates to provide a more robust dataset for genome-wide DNA methylation analysis.

## Methods
### Soybean plant material and growing conditions
Three types of wild soybean (*G. soja*) and eight cultivars, including ZYD00006 (ZYD6), Baiyangdian (Bai YD) and Dongying (Dong Y); Williams 82 (W82), Jack, Dongnong 50 (DN50), Dongnong 60 (DN60), Dongnong 42 (DN42), Hefeng 51 (HF51), Hefeng 55 (HF55) and Suinong 14 (SN14), were used in this study. All soybean materials used in this study were obtained from the Chinese Academy of Agricultural Sciences (CAAS), Northeast Forestry University (NEFU), and Northeast Agricultural University (NEAU). Six progeny populations, R183, R147, R102, R191, R178 and R84 of CSSL derived from SN14 × ZYD6, were obtained from Northeast Agricultural University[52,53]. The plant materials were placed in the experimental field or growth chamber (Percival, USA) of Northeast Forestry University and grown with 11 h of light, at temperatures of 28 °C during the day and 23 °C at night. The growth chamber was equipped with a Philips GreenPower LED Toplighting light (high output system, 410 micromoles/sec).

## Identification of DME and construction of the phylogenetic tree

The DME amino acid sequences of 9 species of plants, including soybean (*Glycine max*), alfalfa (*Medicago truncatula*), kidney bean (*Phaseolus vulgaris*), red clover (*Trifolium pratense*), *Arabidopsis thaliana*, potato (*Solanum tuberosum*), tomato (*Solanum lycopersicum*), maize (Zea mays), rice (*Oryza sativa*) and *Marchantia polymorpha*. These sequences were selected to perform a maximum likelihood analysis using MEGA 10.0 software. First, selected codons were aligned using the following settings: ClustalW algorithm, pairwise alignment, gap opening penalty of 10.00, gap extension penalty of 0.10. For multiple alignment, the settings were as follows: gap opening penalty of 10.00, gap extension penalty of 0.20. Then, the default parameter settings for the remaining parameters in the MEGA X built-in program, "find best protein models ML", were selected to test the most appropriate parameters. The most appropriate parameters were "model/method" choice "Jones-Taylor-Thornton (JTT) model". For "rates among site", choose "gamma distributed with invariant sites (G + I)"; for "gaps/missing data treatment", choose "partial deletion"; for site coverage cut-off (%), choose 50%. The robustness of each node in the tree was determined using 1000 bootstrap replicates. Then, the default parameter settings were selected for the remaining parameters[54].

## Generation and identification of *GmDMEa* mutants

We manually designed a target for gRNA construction. The Phytozome v13.1 (https://phytozome-next.jgi.doe.gov) website was used to identify the soybean *GmDMEa* (*Glyma.10G202200*) sequence, and Geneious software (v8.0) was used to identify the gene. The subregion sequences were aligned, and a single guide RNA (gRNA, 5'- AAGGCCACAGAGATGCTGGA-3') was designed in this region. The sgRNA was located in the first exon, from 592-611 bp (from the start codon ATG). Both forward and reverse primers for the sgRNA were denatured (slow cooling from 95 °C to 16 °C at 0.1 °C/s) before the annealing reaction. Two microlitres of the annealing product was ligated to the pCBSG04 vector at 25 °C for 30 min. The reaction system contained the following components: 1 μl 10x buffer (NEB), 1 μl BsaI-digested pCBSG04 plasmid, 2 μl annealed product (1/100), 0.5 μl T4 ligase (NEB), 0.1 μl T4 PNK (NEB), and 5.4 μl ddH₂O. The restriction endonuclease *Lg*UI was inserted into the pSC1 plasmid vector, and the recombinant vector contained the *Cas9* gene driven by the *GmUbi3* promoter, the sgRNA driven by the *GmU6* promoter, and the bar gene driven by the *CaMV35S* promoter. The recombinant vector was amplified and propagated in competent *E. coli*. After confirming that the base sequence mutation was correct, it was transformed into EHA105 Agrobacterium by repeated freezing and thawing. Mutants were obtained via the Agrobacterium-mediated transformation of soybean cotyledonary nodes. Plants regenerated from a glufosinate-resistant callus of soybean cultivar DN50 were used for agrobacterium-mediated tissue culture transformation. The transformation of soybean by *Agrobacterium tumefaciens* EHA105 has been described previously[55]. All positive lines in generations T₀ and T₁ were confirmed by PCR amplification and Sanger sequencing. After screening with 8 mg/L glufosinate ammonium, genomic DNA was extracted from glufosinate-resistant plants by the CTAB method, and genomic fragments corresponding to sgRNA and Cas9 were amplified (primers: GmDMEa-F and GmDMEa-R, Supplementary Data 14). CAPS markers validated all *gmdmea* T₂ mutants. *Sfa*NI NEB (*gmdmea-1* to *5*) and *Xcm*I NEB (*gmdmea-6, gmdmea-7*) were used to digest PCR products of DN50 and mutants.

## DNA extraction and WGBS

Each sample of mature seeds was broken into small pieces in a grinding bowl, covered with liquid nitrogen, and ground thoroughly to obtain a powder. DNA was then extracted using the Super Plant Genomic DNA Kit (TIANGEN Biotech, Beijing, China). After desalting, the ZYMO EZD DNA methylation-Gold kit (Sigma) was used for bisulphite treatment of the extracted DNA. DNA fragments were selected for size using a sizing gel, and PCR amplification was performed to construct the MethylC-seq library. Finally, sequencing was performed on an Illumina HiSeq2000 sequencer (Shanghai, China) in paired-end mode to generate 150-nt long reads with

over 30× sequencing depth of the soybean genome for each sample. Each biological replicate was sequenced once for the dry seeds of each genotype.

## Data analysis of soybean DNA methylation

Adapter sequences were removed from the raw data from the Illumina HiSeq platform by fastp version 0.19.1 (https://github.com/OpenGene/fastp). Quality control was performed by FastQC version 0.11.9 (https://github.com/s-andrews/FastQC) following standard rules to filter the raw data and to obtain "clean reads", where the N base content was <10%, the base quality at the 3' end >20, and the length of reads >70. The clean reads were mapped to the soybean reference genome (W82.a2.v1)[56] through Bsmap version 3.4.2 (https://github.com/genome-vendor/bsmap). Fisher's exact test and the sliding window algorithm were used for the genome-wide analysis of differentially methylated regions (DMRs, *p* < 0.05) according to previous reported parameters[57]. The fold change of the DMR methylation level had to be greater than 2; when different regions were separated by a distance of less than 200 bp, they were defined as one DMR. The regions were merged to obtain the final DMRs. The Bonferroni method was applied to calculate the false discovery rate (FDR) to correct the P value for multiple testing, and an FDR < 0.05 was therefore considered statistically significant. More details show in Supplementary Data 12.

## RNA extraction and RNA-seq

In our transcriptome sequencing experiment, we conducted three biological replicates for each soybean germplasm sample. The samples for each germplasm were independently collected and processed, with each batch undergoing its own RNA extraction and sequencing. The dry seeds were pulverized into powder under liquid nitrogen using a mortar and pestle. RNA extractions were performed with the RNAprep Pure Plant Plus Kit (TIANGEN Biotech, Beijing, China). Sequencing of the RNA libraries was executed on the Illumina HiSeq 4000 platform, producing 150-bp paired-end reads.

The raw reads were processed with Fastp to trim adaptor sequences and filter out low-quality reads. The resulting high-quality reads were then aligned to the soybean reference genome using Hisat2 with default parameters[58,59]. SAMtools was employed to extract uniquely mapped reads to individual genes[60]. The mRNA abundances were quantified and normalized to c.p.m. and FPKM values using StringTie[61]. Differential gene expression analysis across samples was carried out using the DESeq2 package in R, following the advice of its manual[62]. The GO enrichment analysis utilized background data from SoyBase.org[63].

The sequencing output generated total reads ranging from approximately 36 to 84 million per sample, with an average Q20 rate exceeding 95% and an average Q30 rate above 91%. The GC content of reads averaged between 45.43% and 48.08%, while the mapping rates to the reference genome averaged over 90%, more details show in Supplementary Data 13.

## Real-time quantitative PCR

After RNA extraction, cDNA was synthesized using the PrimeScript™ RT Reagent Kit (TAKARA), and a reverse transcription reaction was carried out. *GmTUB* (NM_001250372.2) was used as the internal reference gene, consistent with previous literature[64]. The expression of target genes was measured by qRT-PCR (primer sequences, see Data 9), for which the reaction system and conditions were set based on the instructions of SYBR Premix Ex Taq™II (TAKARA). The transcript abundance of each gene was calculated using the $2^{-(-\Delta\Delta Ct)}$ CT method[65]. All qRT-PCR experiments were performed with three independent technical replicates. Primer design was performed online at http://www.quantprime.de/ [67](Supplementary Data 14).

## McrBC-qPCR

McrBC digestion of DNA samples of all dry soybean seeds was carried out in a volume of 20 μL, containing 2 μL 10 × NEB buffer, 1 μL 2 mg/mL BSA, 2 μL 10 mM GTP, 1 μL McrBC enzyme (NEB), and 400 ng DNA, brought to the final volume of 20 μL using ddH₂O. The reaction was incubated at 37 °C

for 16 hours. A sample (10-fold diluted) of the digested DNA was used as a template for real-time quantitative PCR using the same machines and procedure described in the qRT-PCR section. Primer design for McrBC-qPCR followed the same procedures described above (Supplementary Data 14).

## Measurement of oil, protein and ABA contents in DN50 and *gmdmea-3* seeds

The protein and oil contents were determined using an Infratec 1241 Seed Analyser (Foss Tecator, Höganäs, Sweden) with a path length of 18 mm. We used the enzyme immunoreaction method to determine the ABA content of seeds in different stages of maturation. In the considered developmental stages, the pods were removed, and only the seeds were selected for analysis. Immediately after sampling, they were frozen in liquid nitrogen and randomly stored in a freezer at −80 °C degrees. The seeds were gently crushed with a mortar and pestle to obtain approximately 20 mg fine powder, which was then transferred to a 1.5 mL Eppendorf tube; 1 mL of extraction buffer (90% (v/v) methanol; 200 mg/L sodium dithiocarbamate trihydrate) was then added, and mixing was performed by careful pipetting. The EP tubes were incubated overnight at 4 °C in the dark with shaking. Centrifugation was conducted at 8000 x g for 10 min at 4 °C. The supernatant was transferred to a new, prechilled 1.5 mL Eppendorf tube, and the ABA concentration was determined using the Phytodetek ABA enzyme immunoassay kit (Agdia). Colour absorbance was measured at 405 nm using a Spectramax 250 microplate reader.

## Frozen sectioning and cell measurements

The soybean seeds were fixed overnight with FAA (100 mL FAA = 89 ml 50% alcohol + 6 mL glacial acetic acid + 5 mL formalin) and were soaked and saturated with 15% glycerin. The samples were then removed and placed in a 20 °C incubator for 3 h, embedded in Tissue-Tek ® O.C.T. Compound, frozen for 20 min and then sliced at a 5 μm thickness. The sections were stained with 0.1% toluidine blue and observed under a light microscope. CaseViewer software was used to save the image. ImageJ software was used for cell counting. We selected six sections (each from an individual seed) at the same location, and five square areas of fixed size (200 μm * 200 μm) were randomly examined in each section. The number of cells in each area was manually counted. The average cell size was calculated as the area/number of cells of each square area based on 30 replicates.

## Phenotypic measurements

The length, width and thickness of dry and fresh seeds were measured with electronic vernier callipers, and at least 10 seeds were selected ($n > 10$). Plant height was measured with a common ruler at different growth stages (R1: One flower open on any node of the main stem; R2: flowers open on either of the two uppermost nodes of the main stem with fully grown leaves; R4: 2 cm long pods on any of the uppermost four nodes of the main stem bearing fully grown leaves; R6: This stage also is known as the "green bean" stage or beginning full-seed stage, and total pod weight will peak during this stage; R8: On the soybean plant, 95% of the pods have reached their mature color.). The number of pods and branches was counted manually at the R8 stage.

## Statistics and reproducibility

Detailed information about the experimental design and statistical methods used in different data analyses in this study is provided in the respective sections of the Results and Methods. DEGs were assessed using Student's *t*-test, while DMRs were analyzed using a hypergeometric test. GO enrichment assessment was conducted using the GO enrichment plugin in TBtools. Statistical analyses, including Student's *t*-test and Wilcoxon test, were performed using GraphPad Software, and bar charts and volcano plots were generated accordingly.

## Data availability

The sequencing raw data for all soybean samples, including whole-genome bisulfite sequencing (WGBS) for DNA methylation analysis and RNA sequencing (RNA-seq) for transcriptomic analysis, are available in the National Center for Biotechnology Information (NCBI) Sequence Read Archive (SRA), under the project accession number PRJNA1039334. The source data underlying Fig. 1c–e, g, h can be found in Supplementary Data 1, 2, 3; Fig. 4c, g can be found in Supplementary Data 7–9; Fig. 5a, b can be found in Supplementary Data 11. All other data related to this study can also be available upon reasonable request to the corresponding or 1st author.

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

## Acknowledgements
We are grateful to Dr. Chentao Lin from the Center for Basic Forestry Research, College of Forestry, Northeast Forestry University kindly provide suggestion and discussion. We are grateful to Dr. Tianfu Han from the Institute of Crop Sciences, Chinese Academy of Agricultural Sciences kindly provide soybean seeds, Baiyangdian, Dongying, Hefeng 51 and HeFeng 55. We would like to thank Dr. Zhimin Zheng from Northeast Forestry University for kindly provide the soybean seeds of Williams 82 and Jack. This work was supported by the National Natural Science Foundation of China (31801444, 31871220); Heilongjiang Provincial Natural Science Foundation of China (LH2021C005); the National Key R&D Program of China during the 14th Five-year Plan Period (2021YFD2200103, 2021YFD2200304); the National Nonprofit Institute Research Grant of the Chinese Academy of Forestry (CAFYBB2019ZY003).

## Author contributions
L.N.X. and Q.Z.Z. conceived and designed the research. W.P.W., C.Y.L[1]., J.W., Z.Z.L., performed the experiments and analysed the data. S.W.S and T.X.Z. performed the bioinformatics analysis. Z.F.J. has provided Dongnong 50 and Dongnong 60 soybean seeds along with field planting management guidance. Q.S.C. and C.Y.L[5]. provided the CSSL seeds. W.P.W., L.N.X., S.A, Z.H.Z, and T.X.Z. wrote the manuscript. All authors discussed the results, revised the draft manuscript, and read and approved the final manuscript.

## Competing interests
The authors declare no competing interests.
