## [Peer review file · Communications Biology]

Reviewers' comments:

Reviewer #1 (Remarks to the Author):

The manuscript describes knocking down DMEa gene in Glycine by the CRISPR–Cas9 method in DN50 (a cultivar with small seeds and a dense-planting architecture). The authors performed methylome and gene expression analysis in large seeded and small seeded cultivars and observed higher CHH methylation in large seeded cultivars associated with MuDR TEs, along with lower DME expression. The authors also show that this CHH methylation change is inheritable by single backcrossing population of small seeded and large seeded cultivars. While the results of the study are novel and interesting, it is important to explain few of the points in details for better understanding and replicability of the study:

- 1) The methylome analysis details are missing (number of replicates used for each genotype, as well as mapping and other stats for the sequencing data).
- 2) Explanation about the product sizes of CASP marker in knockout lines along with its specificity wrt DEMA and DMEb is needed. The expression of DMEa needs to be reported in these mutants as well. Also, show the expression of DMEb in large seeded and small seeded soybean plants.
- 3) The authors may try some more statistical tests as the protein content of knockout seeds seems to be different than DN50 seeds (Figure 2e).
- 4) Mention the number of DME regulated gene with upstream TE and whole genome genes with upstream TEs as the number of hyper DMR genes is only around 95 in Figure 4g and h.
- 5) What is the pattern of expression of other demethylases especially DMEb and RDR2 components in small seeded and large seeded CSSL progenies.
- 6) Data availability of RNA seq methylation data should be clearly specified.
- 7) There seems to be some mistake in supplementary tables S3 and S4 as it is not clear where this data has been used.
- 8) No details about bisulfite sequencing and data analysis is provided in the supplementary tables as well.

Reviewer #2 (Remarks to the Author):

Wang et al. found the negative correlation between expression of GmDMEa, a glycosylase responsible for initial active DNA demethylation, and seed size of 11 soybean accessions through combined transcriptome and methylome analysis, and confirmed the suppressed role of GmDME in seed size using the CRISPR–Cas9 generated GmDMEa alleles mutants. The authors also revealed that GmDMEa regulates gene expression by affecting the methylation levels of TEs, especially MuDRs, and decreased expression of ABA-responsive genes and transcription factors could contribute to the bigger seeds of the GmDMEa alleles mutants. The results provide an important and potential candidate gene for soybean seed genetic improvement and high yield. The manuscript is well organized and potentially interesting for DNA methylation and crop seed development field. However, at this point I have some concerns.

Here are my major comments.

1. The abstract should be refined to make it more clear and concise.

2.The expression of most genes might have been possibly suppressed in the dry seeds, while the authors performed RNA-seq and methylome analysis using dry mature seeds in the present study. Could the authors explain the possible reason?

3. The paragraph between Line267 and Line277 was confusing. Please describe the section more clear.

4.The authors should give more detailed description for all figure legends.

There are a list of more detailed suggestions for the authors to consider in their revision.

1. Line90, "... Active-DNA demethylation ..." should be "... Active DNA demethylation ..."

2. Line121-122, "...There were 490 upregulated genes and 768 downregulated genes in the large-seed group compared to the small-seed group...", Are the 490 and 768 genes shared by all the big or small seed cultivars? it's not clear.

3. Line162-164, "...Two members of the demethylase family in soybean, GmDMEa (Glyma.10g202200) and GmDMEb (Glyma.20g188300), with GmDMEa showing higher homology to AtDME (AT5G04560) and MtrDME (Medtr1g492760)..."

The sentence may be not complete.

4.Line213, "... architecture was observed in different growth stages from R1 to R8 or in the harvestable stage ...". It should be "... at different growth stages from R1 to R8 or at the harvestable stage ..."

5.Line314/324, "... CHH-hyperassociated genes ...", could be "... CHH-hyper associated genes ...".

6.The data is available and the method is well described, while the statistics need to be more clear for all figure and supplementary figure panels.

7.Fig 5c should be improved, ie, the left label DN50 does not match the right diagram well.

Reviewer #3 (Remarks to the Author):

The manuscript by Wanpeng Wang et al. reported that soybean demethylase GmDMEa removed CHH methylation on AT-rich TE through multi-omics, genetic and other experiments, and promoted the expression of seed development-related genes, thereby resulting in smaller seeds in soybean. Authors revealed an epigenetic regulatory mechanism governing seed size in soybean. This article is original and comprehensive. It will be easier for readers to understand it if the following comments are improved. Here are my suggestions for the manuscript.

Major comments:

1. The author conducted RNA-seq and WGBS of 11 soybean germplasm dry seeds, including large-seed (five) and small-seed (six) groups and analyzed them in Figure 1. Please provide the ID of the

sequencing data deposited into the NCBI Sequence Read Archive. If it is published data, please supplement the data source in RESULTS and DATA AVAILABILITY STATEMENT. The RNA-seq replicates of each sample should be presented in the section. The authors had better utilize PCA to show the clustering of the large and small seed according to RNA-seq data. Please make it clear in the text for the reader to understand easily.

2. The authors mentioned that GmDMEa removed CHH methylation on the AT-rich TE in the promoter region of the ABA-responsive gene, which is a very good conclusion. I think heatmaps or some other ways to show the changes in transcription and methylation levels of ABA responsive genes in wild type and mutant.

Minor comments:

1. In line 143: "semidwarfing" should be "semi-dwarfing" or "semi dwarfing".
2. Fig 1e, f: Please label the significant with P value, but not *.
3. Fig 1h and line 259: In the MS, the authors used 10 kb upstream and downstream to investigate DNA methylation alteration. while in previous analysis on DNA methylation in soybean, peanut and cotton (Li et al., 2023; Ma et al., 2018; Rambani et al., 2020), it is common to exhibit DNA methylation changes on upstream and downstream 2 kb of genes or TEs. Please clarify it.
Li, Z., Liu, Q., Zhao, K., Cao, D., Cao, Z., Zhao, K., . . . Yin, D. (2023). Dynamic DNA methylation modification in peanut seed development. *iScience*, 26 (7), 107062. doi:10.1016/j.isci.2023.107062
Ma, Y., Min, L., Wang, M., Wang, C., Zhao, Y., Li, Y., . . . Zhang, X. (2018). Disrupted Genome Methylation in Response to High Temperature Has Distinct Affects on Microspore Abortion and Anther Indehiscence. *Plant Cell*, 30 (7), 1387 1403. doi:10.1105/tpc.18.00074
Rambani, A., Pantalone, V., Yang, S., Rice, J. H., Song, Q., Mazarei, M., . . . Hewezi, T. (2020). Identification of introduced and stably inherited DNA methylation variants in soybean associated with soybean cyst nematode parasitism. *New Phytol*, 227 (1), 168 184. doi:10.1111/nph.16511
4. Line 213: Please provide more detail information about "R1, R8".
5. Fig4a: Please supplement the changes in the distribution of CG and CHG across the whole genome.
6. Figure legends of Fig 5e are missing.

Dear Reviewers,

We would like to express our sincere gratitude for your valuable time and thoughtful comments on our manuscript titled **DNA demethylase suppresses seed size by decreasing the DNA methylation of AT-rich transposable elements in soybean, tracking number: COMMSBIO-23-3589A**. Your expertise and insights have greatly helped us improve the quality and clarity of our work. In this response, we address each of your comments and suggestions in detail, providing clarifications, revisions, and additional information as requested. We genuinely appreciate your constructive feedback and the opportunity to enhance our research.

Once again, thank you for your valuable input, and we hope that the revised version of the manuscript adequately addresses your concerns.

Sincerely,

Linan Xie

Reviewers' comments:

Reviewer #1 (Remarks to the Author):

The manuscript describes knocking down DMEa gene in Glycine by the CRISPR–Cas9 method in DN50 (a cultivar with small seeds and a dense-planting architecture). The authors performed methylome and gene expression analysis in large seeded and small seeded cultivars and observed higher CHH methylation in large seeded cultivars associated with MuDR TEs, along with lower DME expression. The authors also show that this CHH methylation change is inheritable by sung backcrossing population of small seeded and large seeded cultivars. While the results of the study are novel and interesting, it is important to explain few of the points in details for better understanding and replicability of the study:

1) The methylome analysis details are missing (number of replicates used for each genotype, as well as mapping and other stats for the sequencing data).

Thank you for your valuable feedback regarding our manuscript. We have carefully considered your comments about the methylome analysis details and have made the following revisions to address your concerns:

1. We have clarified the number of biological replicates used for each genotype by adding the following sentence to the methods section: “Each biological replicate was sequenced once for the dry seeds of each genotype.” This ensures transparency about the repetition and consistency of our experimental approach.

2. We have provided additional details on the sequencing data, including mapping statistics and other relevant metrics. This includes specifying the sequencing depth and read length, which are now clearly stated as: “Sequencing was performed on an Illumina HiSeq2000 sequencer in paired-end mode to generate 150-nt long reads with over 30× sequencing depth of the soybean genome for each sample.”

3. To ensure completeness, we have also included a supplementary table (**Supplementary Table 12**, here show as **Table R1**) that summarizes the sequencing statistics for each sample, providing a comprehensive overview of the data generated from our study.

We believe that these amendments have strengthened the manuscript by providing a clearer understanding of our experimental procedures and the robustness of our dataset. We appreciate the opportunity to enhance the quality of our work with your insights.

Sample	Total Reads	Total Bases	Total Reads with Ns	N Reads (%)	Q30(%)	GC(%)	BS conversion rate(%)	Uniquely Mapped Read Pairs	Mapping Efficiency (%)	Error Rate (%)	Total Methylated Cytosines (%)
gmdmea-3	196,007,822	29,128,582,384	30,549	0.02	91.16	19	99.53	56,480,707	57.6	0.0278	14.74
Dongnong50	206,402,302	30,620,162,433	88,071	0.04	90.25	19.07	99.48	58,675,662	56.9	0.0289	14.24
Hefeng55	266,966,392	38,238,975,520	7,242,059	2.71	85.2	18.97	99.42	54,015,115	40.5	0.0554	27.59
ZYD00006	221,073,238	31,777,982,898	4,908,796	2.22	85.68	18.36	99.44	38,087,360	34.5	0.0541	26.86
W82	237,346,472	33,830,019,784	1,970,085	0.83	86.91	19.81	99.21	66,893,850	56.4	0.0509	24.21
Jack	260,339,776	37,163,860,984	2,161,443	0.83	86.8	18.48	99.49	63,275,266	48.6	0.0511	22.71
Dongnong60	208,502,148	29,750,258,470	5,497,275	2.64	86.79	19.19	99.47	45,204,130	43.4	0.0511	24.36
Dongying	218,531,484	31,143,380,350	6,216,888	2.84	86.27	19.23	99.32	37,442,642	34.3	0.0524	22.28
Baiyangdian	207,733,450	29,526,958,546	5,947,116	2.86	85.99	19.28	99.44	34,152,934	32.9	0.0532	23.64
Suinong14	209,987,402	30,274,259,770	6,533,537	3.11	85.81	18.87	99.4	45,700,279	43.5	0.0537	23.4
Hefeng51	211,960,090	30,513,268,130	6,430,297	3.03	85.22	18.45	99.29	46,427,509	43.8	0.0554	27.99
Dongnong42	246,271,614	35,247,825,578	6,744,163	2.74	85.32	19.17	99.47	46,019,759	37.4	0.0551	24.04

Table R1. Summary of seed methylomes from wild type (Dongnong50), mutant (*gmdmea-3*) and other soybean germplasms, related to Figure 2.

2) Explanation a out the product sizes of CAPS marker in knockout lines along with its specificity wrt DMEa and DMEb is needed. The expression of DMEa needs to reported in these mutants as well. Also, show the expression of DMEb in large seeded and small seeded soybean plants.

Thank you for your inquiry about the CAPS marker specificity in our study. To address the differentiation between *GmDMEa* and *GmDMEb* genes, we performed a thorough

sequence alignment using the DNAMAN software, which allowed us to identify the specific cleavage sites (CAPS). To distinguish between *GmDMEa* and *GmDMEb*, we designed primers specifically targeting the DNA sequence of *GmDMEa* (Forward primer: CAGAAAGCAACCCAGCGAAG, Reverse primer: TCTTCCTGAGCTGGACCTTC), represented by green arrows and green boxes in our alignment. Upon comparing the genomic sequences of *GmDMEa*, *GmDMEb*, and the *gmdmea-3* mutant, it is evident that the DNA sequences of *GmDMEa* and *GmDMEb* are markedly distinct, confirming the high specificity of our designed primers for *GmDMEa*. In the sgRNA-targeted region (indicated by the red box), the recognition site for the restriction enzyme *SfaI* (GATGC) efficiently cleaves the PCR product of WT-*GmDMEa*. In the *gmdmea-3* mutant, this recognition site is absent, leading to a loss of cleavage in the mutant line. This conclusive difference was utilized to distinguish between the wild-type and knockout alleles in our analysis. **(Figure R1)**

Secondly, regarding the *GmDMEa* expression in the *gmdmea-3* mutants, our CRISPR/Cas9-mediated edits resulted in a premature stop codon, halting the translation process. Despite this truncation at the DNA level, the transcript remains complete, and no change in the transcription level was observed when comparing WT and *gmdmea-3* (as shown by nonsignificant FPKM differences in the bar chart, $p > 0.05$, two-tailed t-test) **(Figure R2)**.

Lastly, in Supplementary Fig 1, we have illustrated the expression levels of *GmDMEa* and *GmDMEb* across different seed sizes through a heatmap. The data revealed no significant expression differences for *GmDMEb*, whereas *GmDMEa* expression exhibited a negative correlation with seed size, prompting us to proceed with gene editing for *GmDMEa* to further investigate its role in seed development **(Figure R3)**.

We trust that these elaborations comprehensively address your concerns and demonstrate the meticulous approach we have taken in our study to validate the specificity of our markers and the integrity of our gene expression data.

gmdmea-3.seqATGGATCAACT	11
WT-GmDMEa.seqATGGATCAACT	11
WT-GmDMEb.seq	TGCTGTCACTGCTAGT TTTACCAACTCTCTCCAATCGGTACCAAAAATGATGGATCAACT	2940
gmdmea-3.seq	CAAAATTTTGTGGACAACCAGTTTTTTTACAATACCAGATTACATAATAGCTGAAAGCACAAAG	71
WT-GmDMEa.seq	CAAAATTTTGTGGACAACCAGTTTTTTTACAATACCAGATTACATAATAGCTGAAAGCACAAAG	71
WT-GmDMEb.seq	CAAAATTTTGTGGACAACCAGTTTTTTTACAATACCAGATTACATAATAGCTGAAAGCACAAAG	3000
gmdmea-3.seq	TCAGGAGAAAGACAAAACAAAACGACTTACTTTTCTTCCACTCAGAATGAACTTAGGAAACA	131
WT-GmDMEa.seq	TCAGGAGAAAGACAAAACAAAACGACTTACTTTTCTTCCACTCAGAATGAACTTAGGAAACA	131
WT-GmDMEb.seq	TCAGGAGAAAGACAAAACAAAACGACTTACTTTTCTTCCACTCAGAATGAACTTAGGAAACA	3060
gmdmea-3.seq	TTTCCGATGGGCTTCTACAGCAAAAATGTTGATTCATCATCTCCAGCAATTTCTACAACATA	191
WT-GmDMEa.seq	TTTCCGATGGGCTTCTACAGCAAAAATGTTGATTCATCATCTCCAGCAATTTCTACAACATA	191
WT-GmDMEb.seq	TTTCCGATGGGCTTCTACAGCAAAAATGTTGATTCATCATCTCCAGCAATTTCTACAACATA	3120
gmdmea-3.seq	TGGGATCAAAAGGGTTCTCACAACATCGTGGCAAGGGAAGTCACTGGGTTTTTCACT	251
WT-GmDMEa.seq	TGGGATCAAAAGGGTTCTCACAACATCGTGGCAAGGGAAGTCACTGGGTTTTTCACT	251
WT-GmDMEb.seq	TGGGATCAAAAGGGTTCTCACAACATCGTGGCAAGGGAAGTCACTGGGTTTTTCACT	3180
gmdmea-3.seq	CAACAAGACACCTGAGCAGAAAGCAACCCACGGAAGAAAGCATAGCCAAAAGTAAITTTAA	311
WT-GmDMEa.seq	CAACAAGACACCTGAGCAGAAAGCAACCCACGGAAGAAAGCATAGCCAAAAGTAAITTTAA	311
WT-GmDMEb.seq	CAACAAGACACCTGAGCAGAAAGCAACCCACGGAAGAAAGCATAGCCAAAAGTAAITTTAA	3240
gmdmea-3.seq	AGAGCCAAAGCCAAAAGAACTCAAAGCCTGCAACTCAAAAACACACAGCTGAAGGCAAA	371
WT-GmDMEa.seq	AGAGCCAAAGCCAAAAGAACTCAAAGCCTGCAACTCAAAAACACACAGCTGAAGGCAAA	371
WT-GmDMEb.seq	AGAGCCAAAGCCAAAAGAACTCAAAGCCTGCAACTCAAAAACACACAGCTGAAGGCAAA	3300
gmdmea-3.seq	TCTGCACAAAAGAGGAAATGTGCCAAAACGCAAGCAACACACAGACCATTTAT	431
WT-GmDMEa.seq	TCTGCACAAAAGAGGAAATGTGCCAAAACGCAAGCAACACACAGACCATTTAT	431
WT-GmDMEb.seq	TCTGCACAAAAGAGGAAATGTGCCAAAACGCAAGCAACACACAGACCATTTAT	3360
gmdmea-3.seq	AGAAAGAAAGTGTGATTTCTATTGTTGCAACTAAAAAATCCTGCAGAAAGGCTCTAAATTT	491
WT-GmDMEa.seq	AGAAAGAAAGTGTGATTTCTATTGTTGCAACTAAAAAATCCTGCAGAAAGGCTCTAAATTT	491
WT-GmDMEb.seq	AGAAAGAAAGTGTGATTTCTATTGTTGCAACTAAAAAATCCTGCAGAAAGGCTCTAAATTT	3420
gmdmea-3.seq	TGACTTGGAAACATAAATAATATGCAAGCCAGAGCACATAGGTTGCCAGCAGGAGATCAA	551
WT-GmDMEa.seq	TGACTTGGAAACATAAATAATATGCAAGCCAGAGCACATAGGTTGCCAGCAGGAGATCAA	551
WT-GmDMEb.seq	TGACTTGGAAACATAAATAATATGCAAGCCAGAGCACATAGGTTGCCAGCAGGAGATCAA	3480
gmdmea-3.seq	CCATAGAAATGAGAAAGCTTTTCAATACCACTTTCAGACCACAAAGGCCACAGCATTTGG	608
WT-GmDMEa.seq	CCATAGAAATGAGAAAGCTTTTCAATACCACTTTCAGACCACAAAGGCCACAGCATTTGG	610
WT-GmDMEb.seq	CCATAGAAATGAGAAAGCTTTTCAATCTAAATTTTGAATGCAACATAAATAATATGCAAG	3540
gmdmea-3.seq	ATGGAGCAACTATGACCTATGCTAAAACTCAGCTTTGCTCATCAG...CTCATGGGAT	664
WT-GmDMEa.seq	ATGGAGCAACTATGACCTATGCTAAAACTCAGCTTTGCTCATCAG...CTCATGGGAT	666
WT-GmDMEb.seq	CCAAAGCACAAATAAGTGGCCAGAGGATGACCATAGCAATGACAAAGCTTTCAATAC	3600
gmdmea-3.seq	GAACTTACACTAGAAAACCAACAAACCAAGAAACACAGAAATATACACTTCTACTGCA	723
WT-GmDMEa.seq	GAACTTACACTAGAAAACCAACAAACCAAGAAACACAGAAATATACACTTCTACTGCA	725
WT-GmDMEb.seq	CACTTCAACCAAGGCCAAAGAAACCAAAACACAGAAATATACACTTCTACTGCA	3660
gmdmea-3.seq	TGAAAAGCAAGCTAATTAATATTCCTGTGCAAAAGCAAAATCAATCACAGCCTTTGTACGCGAC	783
WT-GmDMEa.seq	TGAAAAGCAAGCTAATTAATATTCCTGTGCAAAAGCAAAATCAATCACAGCCTTTGTACGCGAC	785
WT-GmDMEb.seq	TGAAAAGCAAGCTAATTAATATTCCTGTGCAAAAGCAAAATCAATCACAGCCTTTGTACGCGAC	3720
gmdmea-3.seq	TACACAAGAGCCACAGATAGCAAAATTTCTTCTGCTGAGCAAGGTCAGGCTCAGGAAGA	843
WT-GmDMEa.seq	TACACAAGAGCCACAGATAGCAAAATTTCTTCTGCTGAGCAAGGTCAGGCTCAGGAAGA	845
WT-GmDMEb.seq	TACACAAGAGCCACAGATAGCAAAATTTCTTCTGCTGAGCAAGGTCAGGCTCAGGAAGA	3780
gmdmea-3.seq	TTTCTGATTTGTCTCAGCAAGGAAATTAATGGAATGTAATGCATCAATATATTCAACAAAGCA	903
WT-GmDMEa.seq	TTTCTGATTTGTCTCAGCAAGGAAATTAATGGAATGTAATGCATCAATATATTCAACAAAGCA	905
WT-GmDMEb.seq	TTTCTGATTTGTCTCAGCAAGGAAATTAATGGAATGTAATGCATCAATATATTCAACAAAGCA	3840

Figure R1: Specificity of CAPS marker for *GmDMEa* in *gmdmea-3* mutant

Figure R2: Transcriptional analysis of *GmDMEa* in *gmdmea-3* mutant

Figure R3: Expression patterns of *GmDMEa*(*Glyma.10g202200*) and *GmDMEb*(*Glyma.20g188300*) in relation to seed size

3) The authors may try some more statistical tests as the protein content of knockout seeds seems to be different than DN50 seeds (Figure 2e).

In response to your suggestion to explore additional statistical tests regarding the protein content differences observed in knockout seeds compared to DN50 seeds, we have conducted a Wilcoxon signed-rank test to complement the initial t-tests reported in our manuscript. The results of the Wilcoxon signed-rank test did not indicate a statistically significant difference in the oil content between the *gmdmea-1* and DN50 seeds ($Z = -1.472$, Asymp. Sig. (2-tailed) = 0.141), as well as between *gmdmea-3* and DN50 seeds ($Z = -0.736$, Asymp. Sig. (2-tailed) = 0.462). Similarly, the test did not reveal significant differences in protein content between *gmdmea-1* and DN50 seeds ($Z = -1.572$, Asymp. Sig. (2-tailed) = 0.116), and between *gmdmea-3* and DN50 seeds ($Z = -1.219$, Asymp. Sig. (2-tailed) = 0.223). (**Figure R4, R5**)

These findings are consistent with our initial analysis, suggesting that the observed differences in protein content are not statistically significant. This implies that the variations noted may be due to natural variability rather than a systematic change caused by the knockout. We believe these additional analyses strengthen our manuscript by providing a comprehensive statistical perspective and we have updated the manuscript to reflect these results. Thank you again for your constructive comments, which have undoubtedly added to the depth of our analysis.

Test Statistics^a

	gmdmea_1 - DN50	gmdmea_3 - DN50
Z	-1.472 ^b	-.736 ^b
Asymp. Sig. (2-tailed)	.141	.462

a. Wilcoxon Signed Ranks Test
b. Based on negative ranks.

Figure R4 Wilcoxon signed-rank test of oil content

Test Statistics^a

	gmdmea_1 - DN50	gmdmea_3 - DN50
Z	-1.572 ^b	-1.219 ^b
Asymp. Sig. (2-tailed)	.116	.223

a. Wilcoxon Signed Ranks Test
b. Based on positive ranks.

Figure R5 Wilcoxon signed-rank test of protein content

4) Mention the number of DME regulated gene with upstream TE and whole genome genes with upstream TEs as the number of hyper DMR genes is only around 95 in Figure 4g and h.

Thank you for your comment requesting additional context regarding the DME-regulated genes with upstream transposable elements (TEs) compared to the genome-wide occurrence of genes with upstream TEs.

In our study, we identified that out of 49 downregulated CHH-hyper-DMR genes, 38 genes have upstream TEs, accounting for a total of 119 TEs. Among these, MuDR type elements are the most prevalent, representing 39.66% of the TEs identified. This is illustrative of a significant enrichment compared to the genome-wide distribution, where out of 55,589 genes in the soybean genome, 19,853 genes have upstream TEs, with a total of 160,990 TEs, of which MuDR elements constitute 23.96%. The enrichment of MuDR TEs in the upstream regions of the downregulated CHH-hyper-DMR genes suggests a potential role for these elements in the epigenetic regulation of gene expression. It is well-documented that TEs can influence neighboring gene expression through various mechanisms, including alterations in the chromatin landscape and the provision of novel regulatory sequences.

We have added a detailed explanation to the revised manuscript to further highlight the significance of these findings and underscore the potential impact of TEs on gene expression regulation in the context of DME-mediated demethylation (Revised manuscript line 242-257).

We hope this additional information adequately addresses your query and provides a clearer understanding of the implications of our results.

5) What is the pattern of expression of other demethylases especially DMEb and RDR2 components in small seeded and large seeded CSSL progenies.

Thank you for your question on the expression patterns of *GmDMEb* and the two *GmRDR2* genes in small and large seeded CSSL progenies.

Our analysis within the diverse seed size germplasm groups revealed that *GmDMEa* exhibits a significant correlation with seed size, whereas *GmDMEb* does not show an effect. Given that RDR2 acts as a positive factor in the establishment of DNA methylation, we initially hypothesized it might play a role in seed size. However, our focused research on *GmDMEa* was due to its more pronounced impact on seed size variation.

Regarding *GmDMEb*, based on the transcriptome analysis of the CSSL parents Suinong14 and ZYD00006, we observed no significant differences in expression between these two lines, suggesting that variations in *GmDMEb* expression are not associated with seed size differences in these progenies.

As for the genes associated with the RdDM (RNA-directed DNA methylation) pathway (*GmRDR2*: *Glyma.05G035900*, *Glyma.17G091500*), we found that expression levels were lower in the large-seeded parent Suinong14 compared to the small-seeded parent ZYD00006(**Figure R6**). This pattern is consistent with the expression profile displayed in Supplementary Fig. 1, where *GmRDR2* expression is inversely correlated with seed size. Interestingly, whole-genome methylation data indicate that the level of DNA methylation in the large-seeded parent Suinong14 is higher than in the small-seeded parent ZYD00006 (**Figure R7**).

This observation suggests a negative correlation between *GmRDR2* expression levels and DNA methylation levels. In other words, the reduced expression of *GmRDR2* genes in larger seeds does not lead to a decrease in DNA methylation levels. Consequently, we postulate that *GmRDR2* is not the key determinant affecting seed size in our study.

We appreciate the opportunity to clarify these aspects of our research. Further details on these findings have been incorporated into the manuscript to ensure comprehensive understanding and transparency of our study results.

Thank you again for your constructive comments which have undoubtedly strengthened the manuscript

Figure R6 The FPKM of *GmRDR2s* in the CSSL parents

Figure R7 The methylation levels of TEs in the CSSL parents

6) Data availability of RNA seq methylation data should be clearly specified.

The datasets presented in this study can be found in online repositories. The names of the repository/repositories and accession number(s) can be found below: National Center for Biotechnology Information (NCBI) Sequence Read Archive PRJNA1039334

7) there seems to be some mistake in supplementary tables S3 and S4 as it is not clear where this data has been used.

Thank you for bringing this to our attention. We have carefully reviewed Supplementary

Tables S3 and S4 and have ensured that they now accurately reflect the data discussed in the manuscript. To clarify, the data in Supplementary Table S3 corresponds to the identified differentially methylated regions (DMRs) in the CG, CHG, and CHH contexts for the large and small-seed groups, which is referenced in the section discussing DNA methylation levels across different genomic regions (**Fig. 1f, g**).

In Supplementary Table S4, we present the structural features of the demethylase family members identified in soybean and other species. This table supports the text in the section where we analyze the demethylase family and describe the typical demethylase structure (Supplementary Fig. 3b).

We apologize for any confusion caused and have made the necessary corrections to ensure that the tables are presented clearly and accurately reflect the associated findings in the text. These revisions have been highlighted in the revised Supplementary Tables for your convenience.

We appreciate your diligence in assessing our work and hope that these clarifications address your concerns.

8) No details about bisulfite sequencing and data analysis is provided in the supplementary tables as well.

Thank you for your comment regarding the details of bisulfite sequencing and data analysis. We acknowledge the importance of providing comprehensive methodological information to support our findings.

In response to your request, we have compiled a detailed table that outlines the key metrics of our bisulfite sequencing data, including total reads, base quality scores, GC content, bisulfite conversion rates, unique mapping efficiency, error rates, and total methylated cytosines. This table has been included in the revised supplementary materials (now labeled as Supplementary Table 13/**Table R1**).

We believe this additional information will provide a clear understanding of our bisulfite sequencing approach and the subsequent data analysis procedures. We appreciate your attention to detail and hope that these amendments will adequately address your query. Please do not hesitate to contact us should you require any further information.

Reviewer #2 (Remarks to the Author):

Wang et al. found the negative correlation between expression of *GmDMEa*, a glycosylase responsible for initial active DNA demethylation, and seed size of 11 soybean accessions through combined the transcriptomes and methylomes analyse, and confirmed the suppressed role of GmDME in seed size using the CRISPR–Cas9 generated *GmDMEa* alleles mutants. The authors also revealed that GmDMEa regulates gene expression by affecting the methylation levels of TEs, especially MuDRs, and decreased expression of ABA-responsive genes and transcription factors could contribute to the bigger seeds of the GmDMEa alleles mutants. The results provide an important and potential candidate gene for soybean seed genetic improvement and high yield. The manuscript is well organized and potentially interesting for DNA methylation and crop seed development field.

However, at this point I have some concerns.

Here are my major comments.

1. The abstract should be refined to make it more clear and concise.

Thank you for your suggestion to refine the abstract for clarity and conciseness. We have thoroughly revisited the abstract and streamlined it to better summarize the main findings and significance of our study while ensuring that it remains informative and accessible.

The revised abstract succinctly presents the link between GmDMEa expression and soybean seed size, the effect of GmDMEa mutations on yield and seed composition, and the implications of our results for crop breeding and epigenetic control of complex traits.

We believe that these revisions have improved the clarity of the abstract, and we hope that it now meets your expectations. Please find the revised abstract included in our resubmission. We are grateful for your constructive feedback, which has undoubtedly enhanced the manuscript

New abstract:

Soybean seed size is a complex trait regulated by multiple genetic and epigenetic factors. We demonstrate that the DNA demethylase GmDMEa, homologous to DEMETER (DME), is inversely associated with seed size in soybeans. Using CRISPR-Cas9, we generated *GmDMEa* mutants in the small-seed cultivar DN50 and observed an increase in seed size and single plant yield without affecting overall plant structure or seed composition. Our findings suggest that GmDMEa-mediated demethylation preferentially targets AT-rich transposable elements, leading to the activation of genes and transcription factors associated with abscisic

acid response and seed size reduction. Further, chromosomal substitution lines revealed heritable variation in seed size, highlighting an epigenetic mechanism that can be exploited in breeding programs to enhance seed size without compromising oil or protein content. This study provides insights into the epigenetic control of complex traits, offering a strategy for crop improvement.

Keywords: DNA demethylation, DEMETER, Seed size, Soybean, Transposable elements, ABA response.

2.The expression of most genes might have been possibly suppressed in the dry seeds, while the authors performed RNA-seq and methylome analysis using dry mature seeds in the present study. Could the authors explain the possible reason?

Thank you for your insightful question regarding the use of dry mature seeds for RNA sequencing and methylome analysis. We acknowledge that gene expression levels in mature dry seeds may differ from those in developing seeds, potentially leading to reduced RNA yields and alterations in the gene expression profile. However, we chose to use dry seeds for several reasons that are critical for our study's objectives:

Consistency and Comparability: Dry seeds offer a uniform developmental stage that is crucial for comparing genetic and epigenetic differences between varieties. This consistency is less achievable with developing seeds, where slight differences in developmental stages can lead to significant variations in gene expression and methylation patterns.

Stability: Dry seeds are metabolically inactive, which minimizes changes in gene expression and DNA methylation that might occur during sample handling and processing. This stability ensures the integrity of our data.

Precedent and Validity: Previous studies, such as Chen *et al.* (2018), have demonstrated that the DNA methylation landscape of dry seeds closely mirrors that of seeds at developmental stages, supporting the relevance of our approach. It is also noteworthy that our focus was on stable epigenetic marks that are maintained in dry seeds and inherited through generations, which are critical for understanding the heritable aspects of seed size.

Chen, M. *et al.* Seed genome hypomethylated regions are enriched in transcription factor genes. *Proceedings of the National Academy of Sciences* 115, E8315-E8322 601 (2018).

Organ-Specific Expression Analysis: Our organ-specific expression analysis of GmDMEa revealed that it is specifically expressed in dry seeds, thereby validating the relevance of using dry seeds to study its role in seed size regulation.

We believe that these factors collectively justify the use of dry seeds as experimental material for our study. Moreover, the use of dry seeds aligns with our aim to understand the stable and inheritable epigenetic modifications that contribute to seed size variation. Thank you again for the opportunity to clarify this aspect of our study. We hope that this response addresses your query satisfactorily.

3. The paragraph between Line267 and Line277 was confusing. Please describe the section more clear.

Thank you for your feedback regarding the clarity of the section between lines 267 and 277. We understand that the original description of analyzing differentially expressed genes (DEGs) may have been complex and challenging to follow.

In response to your comment, we have carefully revised this section to improve its clarity and coherence. We have streamlined the text to concisely convey our multi-step comparative analysis, which was aimed at identifying DEGs related to seed size in small-seed soybean cultivars while excluding genes unique to wild soybeans that might confound our findings.

Here is a brief overview of the revisions we have made :

1. We clarified the rationale behind our comparison strategy, emphasizing the need to distinguish between genes contributing to seed size variation and those unique to wild soybean ecotypes.

2. We simplified the explanation of how we isolated DEGs specific to the small seed phenotype by systematically excluding genes from the comparisons among wild soybeans and between wild and cultivated soybeans.

3. We provided a clearer description of how we refined the list of candidate DEGs related to seed size by removing genes identified as non-seed size related from the *gmdmea-3* versus DN50 analysis.

We believe these revisions have significantly improved the readability of the methods

used to identify the DEGs associated with seed size and hope that the changes meet your approval. The revised manuscript now includes these modifications for your review (line 260-274). We are grateful for the opportunity to enhance our manuscript and appreciate the guidance you have provided.

4. The authors should give more detailed description for all figure legends.

Thank you for your valuable feedback concerning the level of detail in our figure legends. In response to your suggestion, we have carefully revised the legends to enhance clarity and understanding. We have specifically expanded the descriptions to include more details, particularly regarding the statistical methods employed in our analysis.

By doing so, we aim to provide a comprehensive context that allows for a more intuitive grasp of the figures and their significance within our work. We believe that these improved legends will greatly facilitate the reader's comprehension of the data and the conclusions drawn from it. We are thankful for your guidance in improving the manuscript and ensuring the thoroughness of our presentation.

There are a list of more detailed suggestions for the authors to consider in their revision.

1. Line90, "...Active-DNA demethylation... " should be "... Active DNA demethylation ... "

Thank you for drawing our attention to the typographical error on Line 90 of our manuscript. We have corrected the phrase from "Active-DNA demethylation" to "Active DNA demethylation" as you suggested.

We appreciate your careful reading of our manuscript and your help in improving the text. The correction has been made in the revised version of the manuscript.

2. Line121-122, "...There were 490 upregulated genes and 768 downregulated genes in the large-seed group compared to the small-seed group... ", Are the 490 and 768 genes shared by all the big or small seed cultivars? it's not clear.

Thank you for your insightful comment concerning the expression of 490 upregulated and 768 downregulated genes in the large-seed group compared to the small-seed group. We

appreciate you highlighting the need for clarity on whether these genes are consistently expressed across all cultivars within each seed size category. In our analysis, the identified genes represent a consensus profile derived from comparing aggregate data across multiple cultivars within the large and small seed groups, each with three biological replicates. Therefore, while individual cultivars may exhibit unique variations in gene expression, the reported upregulated and downregulated genes are those that consistently differentiate the two groups on average, rather than being shared identically by all cultivars in a group.

Our study's objective is to elucidate the general expression trends between the large and small seed groups rather than the specific expression profiles of individual cultivars. By utilizing average expression data from multiple cultivars, we aimed to minimize the impact of individual variation and provide a robust overview of the transcriptional differences associated with seed size.

We have revised the manuscript to more clearly reflect this explanation and ensure that the rationale behind our comparative approach is transparent. We believe that this method allows us to identify meaningful expression patterns that are relevant to the phenotype of interest. Thank you once again for your constructive feedback, which has helped enhance the quality and clarity of our manuscript. We hope that this response satisfactorily addresses your query.

3. Line162-164, "...Two members of the demethylase family in soybean, GmDMEa (Glyma.10g202200) and GmDMEb (Glyma.20g188300), with GmDMEa showing higher homology to AtDME (AT5G04560) and MtrDME (Medtr1g492760)..." The sentence may be not complete.

Thank you for pointing out the issue with the sentence structure on lines 162-164. We agree that the original sentence was incomplete and may have caused confusion.

We have revised the sentence to read: "There are two members of the demethylase family in soybean, GmDMEa (Glyma.10g202200) and GmDMEb (Glyma.20g188300), with GmDMEa showing higher homology to AtDME (AT5G04560) and MtrDME (Medtr1g492760)." This change ensures that the sentence is now complete and the information about the homology of GmDMEa is clearly presented. We appreciate your

attention to detail, which has helped improve the manuscript. The revised sentence has been incorporated into the updated version of the manuscript for your review.

4. Line213, "... architecture was observed in different growth stages from R1 to R8 or in the harvestable stage ...". It should be "... at different growth stages from R1 to R8 or at the harvestable stage ..."

Thank you for your careful review and the suggestion to improve the clarity of the sentence on line 213. As you recommended, we have revised the sentence for greater precision and now it reads: "... at different growth stages from R1 to R8 or at the harvestable stage ..."

We value your contribution to refining our manuscript and are pleased to incorporate your suggested changes. The revised manuscript includes this correction.

5.Line314/324, "... CHH-hyper associated genes ...", could be "... CHH-hyper associated genes ..."

Thank you for your suggestion to improve the clarity in the text regarding "CHH-hyper associated genes." We have implemented the recommended change to ensure accurate terminology is used throughout the manuscript.

Please note that the updated text now appears on lines 310 and 320, reflecting the corrections you advised. We are grateful for your attention to detail and appreciate your assistance in enhancing the quality of our manuscript.

6.The data is available and the method is well described, while the statistics need to be more clear for all figure and supplementary figure panels.

Thank you for your constructive comments regarding the statistical details in our figures and supplementary materials. We have thoroughly reviewed all illustrations and supplementary figure panels to ensure that the descriptions of our statistical methods are clearly presented and accurate.

We have now included a more detailed explanation of the statistical analyses accompanying each figure, ensuring that both the methodology and the significance of results

are transparent and understandable. We believe these revisions will facilitate a better understanding of our data and are grateful for the opportunity to enhance the clarity of our presentation

7.Fig 5c should be improved, ie, the left label DN50 does not match the right diagram well.

Thank you for your attentive analysis of our figures and for pointing out the discrepancy in Figure 5c. We have carefully reviewed the figure and concur that the left label 'DN50' did not correspond appropriately with the right diagram.

In response to this issue, we have conducted a thorough revision of Figure 5c to ensure that the labels accurately match the diagrams. This modification will enhance the figure's clarity and ensure that it effectively conveys the intended information.

We appreciate your guidance on this matter and trust that the improvements made will meet the high standards of the journal. We are grateful for the opportunity to refine our work and thank you once again for your valuable contribution.

Reviewer #3 (Remarks to the Author):

The manuscript by Wanpeng Wang et al. reported that soybean demethylase GmDMEa removed CHH methylation on AT-rich TE through multi-omics, genetic and other experiments, and promoted the expression of seed development-related genes, thereby resulting in smaller seeds in soybean. Authors revealed an epigenetic regulatory mechanism governing seed size in soybean. This article is original and comprehensive. It will be easier for readers to understand it if the following comments are improved. Here are my suggestions for the manuscript.

Major comments:

1. The author conducted RNA-seq and WGBS of 11 soybean germplasm dry seeds, including large-seed (five) and small-seed (six) groups and analyzed them in Figure 1. Please provide the ID of the sequencing data deposited into the NCBI Sequence Read Archive. If it is published data, please supplement the data source in RESULTS and DATA AVAILABILITY STATEMENT. The RNA-seq replicates of each sample should be presented in the section. The authors had better utilize PCA to show the clustering of the large and small seed according to RNA-seq data. Please make it clear in the text for the reader to understand easily.

Thank you for your insightful comment. The RNA-seq and WGBS raw sequencing files of the 11 soybean germplasm and mutants have indeed been uploaded to the NCBI Sequence Read Archive, under the accession number: PRJNA1039334. We apologize for the oversight and have now added this information to the 'RESULTS' section and 'DATA AVAILABILITY STATEMENT' in our manuscript.

We also agree with your suggestion to include the RNA-seq replicates for each sample in the text, and have added detailed descriptions of our RNA-seq replicate experimental design and results in the METHODS section (Line 478-480).

Furthermore, we agree that utilizing Principal Component Analysis (PCA) to illustrate the clustering of the large and small seeds based on RNA-seq data is a valuable suggestion. We have added a PCA plot, as per your suggestion, for easier understanding by the readers. We appreciate your guidance in helping to improve our work.

Figure R8. Principal Component Analysis (PCA) of RNA-seq Data Distinguishes Between Large-Seed and Small-Seed Soybean Germplasm Groups

2. The authors mentioned that GmDMEa removed CHH methylation on the AT-rich TE in the promoter region of the ABA-responsive gene, which is a very good conclusion. I think heatmaps or some other ways to show the changes in transcription and methylation levels of ABA responsive genes in wild type and mutant.

Thank you for your time in reviewing our paper and for your valuable suggestion. We understand your interest in a heatmap representation to visually illustrate the differences in ABA-responsive gene behavior between wild type and mutant types. However, in our study, we have used Integrative Genomics Viewer (IGV) screenshots to display the genomic location information of DNA methylation modifications, gene expression levels, and upstream transposons as this method provides rich and detailed information, and better reflects the complex interactions within the genome.

Besides, we have validated the differences in DNA methylation and gene expression between wild types and mutants by McrBC-qPCR and RT-qPCR experiments, ensuring the accuracy and reliability of our data. While we appreciate and respect your suggestion, we believe our current presentation method is more suited to our study. Nonetheless, if you believe that we need to provide more information or present the data in a different way, we are very open to further discussion and improvement.

Minor comments:

1. In line 143: “semidwarfing” should be “semi-dwarfing” or “semi dwarfing”.

Regarding your note on line 143, we have amended the typographical error. The term “semidwarfing” has been updated to “semi-dwarfing” to ensure correct usage throughout the text.

2. Fig 1e, f: Please label the significant with P value, but not *.

Pertaining to your suggestion for Figure 1e and f, we have replaced the asterisks with the specific P values to accurately represent the statistical significance of our findings. We appreciate your attention to detail and agree that this change improves the clarity of our results presentation.

3. Fig 1h and line 259: In the MS, the authors used 10 kb upstream and downstream to investigate DNA methylation alteration. while in previous analysis on DNA methylation in soybean, peanut and cotton (Li et al., 2023; Ma et al., 2018; Rambani et al., 2020), it is common to exhibit DNA methylation changes on upstream and downstream 2 kb of genes or TEs. Please clarify it.

Li, Z., Liu, Q., Zhao, K., Cao, D., Cao, Z., Zhao, K., . . . Yin, D. (2023). Dynamic DNA methylation modification in peanut seed development. *iScience*, 26 (7), 107062. doi:10.1016/j.isci.2023.107062

Ma, Y., Min, L., Wang, M., Wang, C., Zhao, Y., Li, Y., . . . Zhang, X. (2018). Disrupted Genome Methylation in Response to High Temperature Has Distinct Affects on Microspore Abortion and Anther Indehiscence. *Plant Cell*, 30 (7), 1387 1403. doi:10.1105/tpc.18.00074

Rambani, A., Pantalone, V., Yang, S., Rice, J. H., Song, Q., Mazarei, M., . . . Hewezi, T. (2020). Identification of introduced and stably inherited DNA methylation variants in soybean associated with soybean cyst nematode parasitism. *New Phytol*, 227 (1), 168 184. doi:10.1111/nph.16511

We greatly appreciate the reviewer’s in-depth understanding and valuable feedback on our study. You are correct in pointing out that many previous studies have indeed primarily focused on DNA methylation changes in the 2kb upstream and downstream regions of genes (e.g., Li et al., 2023; Ma et al., 2018; Rambani et al., 2020). This is because these regions often contain important regulatory elements, such as promoters and enhancers, that have a direct impact on gene expression.

However, there is also evidence suggesting that DNA methylation changes in a broader region beyond the transcription start site (TSS) and transcription end site (TES) of genes could also influence gene expression (e.g., Gouil et al., 2016; Niederhuth et al., 2016; Yaari et

al., 2019; Huang et al., 2019; Cao et al., 2020). Considering this, we chose a broader 10kb range to investigate DNA methylation changes, in order to capture all possible methylation events that could affect gene expression. Similarly, this broader range also allows us to more comprehensively study the impact of DNA methylation on transposon activity, as transposons can be inserted into areas of genes far from the TSS and TES. Thank you again for your suggestion, we will clarify the reason for our choice of the 10kb range in the revised manuscript.

Gouil Q, Baulcombe DC (2016) DNA Methylation Signatures of the Plant Chromomethyltransferases. *PLoS Genet* 12(12): e1006526. doi.org/10.1371/journal.pgen.1006526

Niederhuth, C.E., Bewick, A.J., Ji, L. et al. (2016). Widespread natural variation of DNA methylation within angiosperms. *Genome Biol* 17, 194. doi.org/10.1186/s13059-016-1059-0

Yaari, R., Katz, A., Domb, K. et al. (2019). RdDM-independent de novo and heterochromatin DNA methylation by plant CMT and DNMT3 orthologs. *Nat Commun* 10, 1613. doi.org/10.1038/s41467-019-09496-0

Huang, L., Chen, X., Dasgupta, C. (2019). Foetal hypoxia impacts methylome and transcriptome in developmental programming of heart disease. *Cardiovascular research*, 115 (8), 1306-1319. doi.org/10.1093/cvr/cvy277.

Cao P, Li H, Zuo Y and Nashun B (2020) Characterization of DNA Methylation Patterns and Mining of Epigenetic Markers During Genomic Reprogramming in SCNT Embryos. *Front. Cell Dev. Biol.* 8:570107. doi: 10.3389/fcell.2020.570107

4. Line 213: Please provide more detail information about "R1, R8".

Thank you for your question. In our study, "R1" and "R8" refer to the reproductive growth stages of soybeans. Specifically, "R1" denotes the stage at which the soybean begins flowering, marked by the first appearance of open blossoms; "R8" signifies the final stage of soybean growth, where 95% of the pods have changed to their mature brown color. Detailed explanations of these stages can be found in line 197 -199 of the revised manuscript. I hope this answers your question, and I am happy to provide more information if needed.

5. Fig 4a: Please supplement the changes in the distribution of CG and CHG across the whole genome.

Thank you for your insightful suggestion to supplement the changes in the distribution of CG and CHG methylation across the whole genome in Figure 4a.

In the original figure, we primarily focused on depicting the distribution of CHH methylation across the genome, as well as the differences between the wild type (DN50) and the mutant (*gmdmea-3*). We intended to ensure that the circos plot did not become overly

complicated and thus preserve the clarity and accuracy of the data representation.

However, we do agree that including the CG and CHG methylation changes would provide a more comprehensive figure. As such, we have created an additional figure (Supplementary Fig. 7a, here as Figure R9) that includes the CG and CHG methylation status in both the wild type and mutant, which we believe addresses your suggestion. We hope this addition enhances the understanding of our findings and provides more context to our study. Once again, we appreciate your valuable feedback and the opportunity to improve our work.

Figure R9. Genome-wide Distribution of CG, CHG and CHH Methylation in wild type (DN50) and the mutant (*gmdmea-3*)

6. Figure legends of Fig 5e are missing.

We sincerely apologize for the oversight. The labels in Figure 5d were indeed misleading. To clarify, the left part of Fig 5d represents the results of McrBC-RT-qPCR, which are experimental validations of DNA methylation differential regions (DMRs), while the right part represents the results of RT-qPCR, which validate differential gene expression. We have corrected these errors in the manuscript. Thank you for bringing this to our attention.

Reviewers' comments:

Reviewer #1 (Remarks to the Author):

The authors have done significant improvement in the manuscript by addressing the concerns highlighted by reviewers in the rebuttal letter. However, in view of the corrections done few of my concerns about the revised manuscript that needs to be addressed before its consideration for publication:

- 1) The images shown in the rebuttal letter are not incorporated in the manuscript figures or supplementary figures.
- 2) It is also difficult to judge where the revision has been done in the manuscript as the changes are not highlighted in the revised manuscript.
- 3) The authenticity of methylation results in view of single replicate for each genotype is difficult to ascertain. It is evident in the revised version only as earlier the information about sequencing was not available.
- 4) The IGV plots for selected ABA pathway genes show both hypo methylation also, however, the authors have only commented about hyper methylation. Have they checked the combined methylation levels in the this region in small seeded and large seeded cultivars. It is important to explain the MCRBC-qRT-PCR result analysis in view of these highlight the regions tested with MCRBC along with some negative control region.

In my opinion these points needs to be addressed before considering the manuscript for publication.

Reviewer #2 (Remarks to the Author):

Wang et al. found the negative correlation between expression of GmDMEa, a glycosylase responsible for initial active DNA demethylation, and seed size of 11 soybean accessions through combined transcriptomes and methylomes analyse, and confirmed the suppressed role of GmDME in seed size using the CRISPR-Cas9 generated GmDMEa alleles mutants. The results provide an important and potential candidate gene for soybean seed genetic improvement and high yield. The manuscript is well organized and potentially interesting for DNA methylation and crop seed development field.

The authors have addressed my concerns, and improved the manuscript adequately.

Reviewer #3 (Remarks to the Author):

I appreciated that the author have greatly improved their MS, clarified the analytical methods and also made substantial improvements to both the figures. I hope the final manuscript will be published

Referee #1:

1) The images shown in the rebuttal letter are not incorporated in the manuscript figures or supplementary figures.

Thank you very much for pointing out the oversight. The images from the previous version of the rebuttal letter have now been appropriately incorporated into the supplementary figures section. We will ensure that these images are included in the final manuscript to provide readers with a more comprehensive understanding and visual representation of our findings. We appreciate your attention to detail and your valuable suggestions.

2) It is also difficult to judge where the revision has been done in the manuscript as the changes are not highlighted in the revised manuscript.

Thank you for pointing out the lack of clarity in our revision presentation. To address this issue, in the second round of our revised manuscript, we have now highlighted (yellow background) all the changes made, including those from the first round of revisions. We hope these highlighted sections will make it easier for you and other readers to identify the updates and improvements made to the manuscript. We appreciate your valuable feedback.

3) The authenticity of methylation results in view of single replicate for each genotype is difficult to ascertain. It is evident in the revised version only as earlier the information about sequencing was not available.

We are sincerely grateful for your valuable suggestion regarding the rigor of our genome-wide DNA methylation sequencing data. We recognize the importance of biological replicates in omics studies to validate the reliability of the data. However, it is worth noting that several prior studies in the field of plant science have also adopted a single replicate strategy for genome-wide DNA methylation sequencing, particularly when considering the relative stability of DNA methylation (Wang et al., 2024; Javier et al., 2018).

In addition, to further validate our sequencing data, we have conducted McrBC-qRT-PCR experiments. This method is widely recognized and repeatedly used for its accuracy in detecting DNA methylation levels (Ghoshal et al., 2021). Through the experimental validation with this technique, our confidence in the sequencing results has been bolstered.

We appreciate your attention to the precision of our data, and we will continue to strive to ensure the reliability and validity of our findings. Thank you very much for your support of our work.

Wang, S., Wang, M., Ichino, L., Boone, B.A., Zhong, Z., Papareddy, R.K., Lin, E.K., Yun, J., Feng, S., & Jacobsen, S.E (2024). MBD2 couples DNA methylation to transposable element silencing during male gametogenesis. *Nature plants*, 10 (1), 13-24.

Javier Gallego-Bartolomé, Jason Gardiner, Wanlu Liu, Ashot Papikian, Basudev Ghoshal, Hsuan Yu Kuo, Jiannan Zhao, David J. Segal, & Steven E. Jacobsen (2018). Targeted DNA demethylation of the Arabidopsis genome using the human TET1 catalytic domain. *Proceedings*

of the National Academy of Sciences of the United States of America, 115 (9)

Ghoshal, B., Picard, C.L., Vong, B., Feng, S., & Jacobsen, S.E (2021). CRISPR-based targeting of DNA methylation in Arabidopsis thaliana by a bacterial CG-specific DNA methyltransferase. Proceedings of the National Academy of Sciences of the United States of America, 118 (23).

4) The IGV plots for selected ABA pathway genes show both hypo methylation also, however, the authors have only commented about hyper methylation. Have they checked the combined methylation levels in the this region in small seeded and large seeded cultivars. It is important to explain the McrBC-qRT-PCR result analysis in view of these highlight the regions tested with McRBC along with some negative control region.

Thank you for your suggestions. You rightly pointed out the inadequacy in our description of the IGV snapshots concerning the states of hyper and hypo methylation, which may cause confusion. To articulate the experimental outcomes more clearly, we have supplemented the figure legends (Fig 5c) with detailed explanations (line 863-867) to more accurately convey the data presented in the IGV plots. Furthermore, we appreciate your advice on the interpretation of the McrBC-qRT-PCR results. A more thorough discussion has indeed been warranted, and we have revised the relevant sections of the text to provide a more precise description of these findings (line 295-304). Thank you once again for your meticulous review and valuable comments.

Referee #2:

We are very grateful for your positive evaluation and the valuable feedback you have provided. Your affirmative comments serve as a significant encouragement to our team and bolster our confidence in our research. We are pleased to know that our revised manuscript has met your expectations and that you find our findings to be of importance for the field of soybean seed genetic improvement and yield enhancement. We look forward to our research contributing to the fields of DNA methylation and crop seed development, and hope it may inspire future studies. Thank you once again for your support and the time you invested in reviewing our work.

Referee #3:

Thank you very much for your thorough review and appreciation of our revised manuscript. We are honored that you have noticed the improvements we made and provided valuable comments on our analytical methods and figures. We are also looking forward to the possibility of our research being published and hope that it will contribute to the field. Your positive feedback is a great encouragement to our team. We hope to continue to enhance the quality of our work and provide valuable research findings to the scientific community. Thank you again for your valuable time and expert advice.

REVIEWERS' COMMENTS:

Reviewer #1 (Remarks to the Author):

The manuscript describes knocking down DMEa gene in Glycine by the CRISPR-Cas9 method in DN50 (a cultivar with small seeds and a dense-planting architecture). The authors performed the revisions for better understanding and replicability of the study. However, there are some minor corrections and the manuscript can be accepted for publication:

- 1) Please incorporate RNA seq data details (reads and mapping details) as shown in response letter in the manuscript file as well.
- 2) On Page 5, line number 147-149, it will be better if authors justify DMEa and DMEb based on expression difference between small seeded and large seeded rather than similarity of these with Arabidopsis DME (2% difference in protein). The basis of saying DMEa show high homology based on phylogenetic tree is not clear as both proteins are present in same clade in phylogenetic tree.
- 3) On page 6, line 169, The reference should be Sup Fig 4 and not Sup fig3c
- 4) On page 7, line 189 reference to sup Fig 2e is missing. Also the details of statistical tests used for this figure is neither mentioned in figure nor in the methods.
- 5) On page 9, line 254, Supplementary Figure 7b shows AT ratios of different types of TEs with MuDr showing about 0.75 ratio, how does it relate to this number?
- 6) On page 10 and 11, The details about the uses of McRBC is not needed, which regions were chosen for this analysis and their positions on IGV will be a better option.

Dear Editor:

Thank you for your editorial feedback requesting a discussion on the use of a single replicate for genome-wide DNA methylation sequencing in our study. We understand the importance of this issue and have added a section in the “Discussion” part of our manuscript to address this limitation.

We have elaborated on the reasons behind our decision to use a single replicate, citing resource considerations and the precedent set by several studies in the field of plant science (Wang et al., 2024; Javier et al., 2018). Additionally, we have discussed our validation efforts using McrBC-qRT-PCR to reinforce the authenticity of our sequencing results. We recognize that despite this attempt at validation, the lack of biological replicates is a limitation that could impact the generalizability of our findings.

We believe that this added discussion will provide readers with a transparent understanding of the methodological constraints of our study while demonstrating our commitment to data integrity. We are grateful for your support and guidance in improving our manuscript and hope that our revisions meet the journal’s standards.

Thank you once again for your consideration.

REVIEWERS' COMMENTS:

Point-by-point response to Reviewer #1's comments:

The manuscript describes knocking down DMEa gene in Glycine by the CRISPR-Cas9 method in DN50 (a cultivar with small seeds and a dense-planting architecture). The authors performed the revisions for better understanding and replicability of the study. However, there are some minor corrections and the manuscript can be accepted for publication:

1) Please incorporate RNA seq data details (reads and mapping details) as shown in response letter in the manuscript file as well.

Thank you for your valuable suggestion. We have revised the “Materials and Methods” section to include a more detailed description of the transcriptome data analysis. Additionally, we have incorporated the details of the RNA seq data (including reads and mapping details) into the supplementary material as Table 13. We appreciate your constructive comments which have helped to enhance the quality of our manuscript.

2) On Page 5, line number 147-149, it will be better if authors justify DMEa and DMEb based on expression difference between small seeded and large seeded rather than similarity of these with Arabidopsis DME (2% difference in protein). The basis of saying DMEa show high homology based on phylogenetic tree is not clear as both proteins are present in same clade in phylogenetic tree.

Thank you for your insightful suggestion, which has certainly helped us to clarify our conclusions. Following your advice, we have added a more detailed description concerning the comparative expression levels of *GmDMEa* and *GmDMEb* between groups with large and small soybean seeds. This addition has been incorporated into the manuscript from lines 161 to 164.

We agree that justifying the distinction between *GmDMEa* and *GmDMEb* based solely on their protein similarity with *Arabidopsis* DME was not sufficient. Your comment prompted us to re-evaluate our approach and fortify our argument with more direct evidence from our expression data,

which indeed supports the role of *GmDMEa* as a significant epigenetic factor in seed size variation.

We are grateful for your constructive feedback, which we believe has substantially strengthened our manuscript. Thank you once again for your guidance and support in improving our work.

3) On page 6, line 169, The reference should be Sup Fig 4 and not Sup fig3c

We have amended the misreference to the figure, and we appreciate your attentive recommendation.

4) On page 7, line 189 reference to sup Fig 2e is missing. Also the details of statistical tests used for this figure is neither mentioned in figure not in the methods.

Thank you for your prompt. We have added the reference to Fig 2e as you indicated. Additionally, we have supplemented the figure legends with details of the statistical methods employed (lines 825-826).

5) On page 9, line 254, Supplementary Figure 7b shows AT ratios of different types of TEs with MuDr showing about 0.75 ration, how does it relate to this number?

Thank you for your guidance. Due to a clerical mistake on our part, we incorrectly cited Supplementary Figure 7c instead of 7b. This has now been rectified (line 259).

6) On page 10 and 11, The details about the uses of MCrBC is not needed, which regions were choosen for this analysis and their positions on IGV will be a better option.

Thank you for your valuable feedback on our manuscript. We have carefully considered your suggestion and have revised the relevant sections accordingly.

In the revised manuscript, we have omitted the extensive details of the MCrBC-qPCR technique itself and have concentrated on elucidating the specific differentially methylated regions (DMRs) of interest within the promoter regions of the genes *Glyma.06G029100*, *Glyma.20G094500*, and *Glyma.01g015700*. These revisions can be found on line 300-312.

We have used IGV snapshots to pinpoint the exact locations of the DMRs, which also contain MuDR transposable elements, and have selected these particular regions for detailed analysis. The primers employed in both MCrBC-qPCR and RT-qPCR to quantify methylation and validate gene expression are listed in Supplementary Table 13. The outcomes of these analyses, which are depicted in Figures 5d and e, have affirmed the methylation and expression patterns observed, lending further credibility to our findings.

We trust that these amendments have addressed your concerns and have enhanced the clarity and focus of our manuscript's methodology and results sections. We appreciate the opportunity to improve our work and thank you for your guidance.